# LAYOUT-YOUR-3D: CONTROLLABLE AND PRECISE 3D GENERATION WITH 2D BLUEPRINT

**Junwei Zhou**[1]    **Xueting Li**[2]    **Lu Qi**[3,4,*]    **Ming-Hsuan Yang**[5,6]

[1]Huazhong University of Science and Technology    [2]NVIDIA    [3]Wuhan University
[4]Insta360 Research    [5]UC Merced    [6]Yonsei University

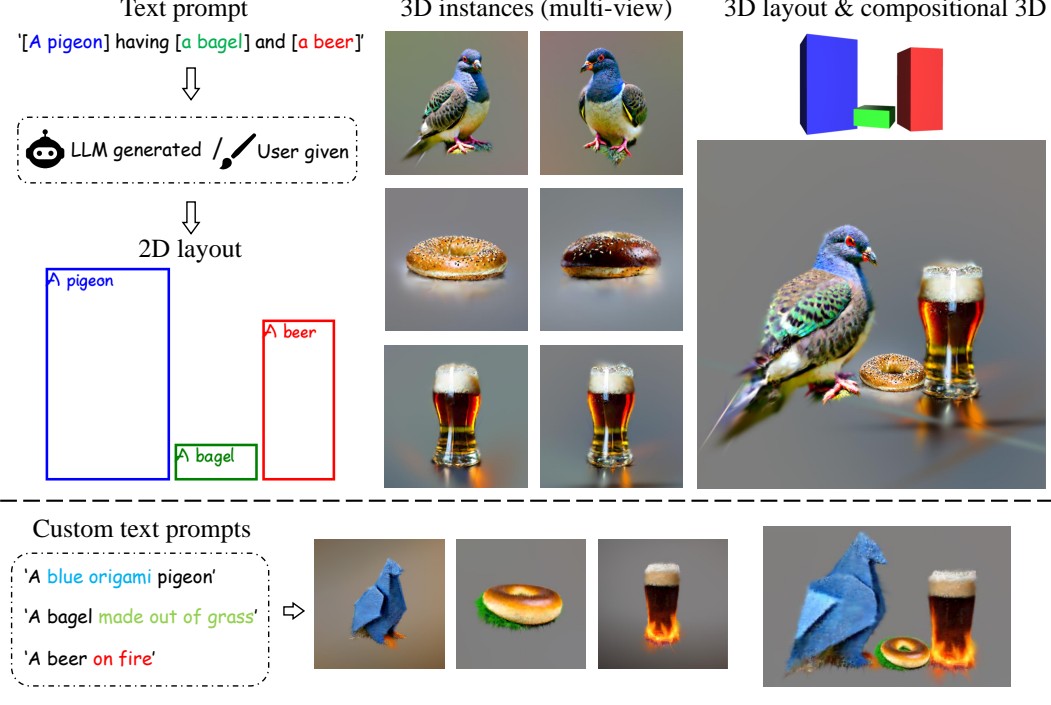

Figure 1: Layout-Your-3D generates high-quality compositional 3D scenes with given 2D layouts (top). Layout-Your-3D further enables editing each instance in the 3D scene with custom text prompts, achieving controllable and precise 3D generation (bottom).

## ABSTRACT

We present Layout-Your-3D, a framework that allows controllable and compositional 3D generation from text prompts. Existing text-to-3D methods often struggle to generate assets with plausible object interactions or require tedious optimization processes. To address these challenges, our approach leverages 2D layouts as a blueprint to facilitate precise and plausible control over 3D generation. Starting with a 2D layout provided by a user or generated from a text description, we first create a coarse 3D scene using a carefully designed initialization process based on efficient reconstruction models. To enforce coherent global 3D layouts and enhance the quality of instance appearances, we propose a collision-aware layout optimization process followed by instance-wise refinement. Experimental results demonstrate that Layout-Your-3D yields more reasonable and visually appealing compositional 3D assets while significantly reducing the time required for each prompt. Additionally, Layout-Your-3D can be easily applicable to downstream tasks, such as 3D editing and object insertion. Video and results can be viewed on our website: https://colezwhy.github.io/layoutyour3d/

---

*Corresponding author

# 1 INTRODUCTION

The recent years have witnessed significant advances in 3D content creation. By leveraging advanced diffusion models through optimization (Poole et al., 2022) or training efficient reconstruction models (Hong et al., 2024), numerous methods successfully synthesize high-quality 3D objects from text prompts or images. These approaches have facilitated various applications such as virtual reality (Chen et al., 2024a; Liu et al., 2024b), robotics (Huang et al., 2023b; Wan et al., 2024), etc.

Nevertheless, existing text-to-3D methods often struggle to synthesize plausible object interactions in compositional 3D asset generation. For instance, given the text prompt "a brown soft with a cushion, a teddy bear and a basketball on it", none of the state-of-the-art methods (Wang et al., 2023; Liang et al., 2024; Yi et al., 2024) shown in Fig. 5 (last row) correctly predict the plausible location and scale of the objects. This is because these optimization-based models create 3D assets by distilling prior knowledge from 2D diffusion models (Rombach et al., 2022). Consequently, they require extensive optimization time for each prompt and suffer from the same issues such as semantic confusion or missing objects as the diffusion models. Meanwhile, feed-forward methods (Tang et al., 2024a) produce 3D shapes instantly but often perform poorly in creating complex compositional cases due to the lack of compositional 3D objects in their training data (Deitke et al., 2023). Furthermore, none of these methods provide any user control over the 3D generation process. These limitations significantly restrict the application of text-to-3D methods in real-world scenarios. Two lines of work aim to resolve these issues and generate plausible compositional 3D assets. First, several methods use various attention mechanisms (Li et al., 2024b) to enhance the alignment of synthesized 3D objects or scenes with compositional text descriptions. However, these methods do not offer controllable signals such as layout. Another line of approaches focuses on text-to-3D scene generation (Zhou et al., 2024b) by first synthesizing each 3D instance within the scene, followed by a global layout refinement through Score Distillation techniques (Poole et al., 2022). While these schemes aim to improve scene coherence, they require significantly more computational resources and generate less realistic 3D results than conventional text-to-3D object synthesis methods.

In this work, we present **Layout-Your-3D**, an efficient text-to-3D generation method for compositional 3D scene synthesis with precise control. Specifically, given a text prompt describing few objects and their spatial relationships, we aim to generate these objects and their natural interactions. We note that this is similar to the setting in (Sun et al., 2024) but differs from (Zhou et al., 2024b; Li et al., 2024a; Gao et al., 2024), which focuses on synthesizing complete indoor/outdoor 3D scenes. Our key idea is to utilize a 2D layout as guidance to produce target reference images for 3D scene generation, which not only allows precise user control but also produces more plausible object interactions. We start by collecting a user-provided or LLM-generated 2D layout and predict a reference image depicting the desired instances and their interactions. Using this reference, we propose a carefully designed process to create coarse 3D scenes with roughly reasonable layouts using reconstruction models (Tang et al., 2024a). While these feed-forward reconstruction models are computationally lightweight, they often yield inferior 3D geometry and appearance. To address these limitations, we introduce a two-step disentangled refinement stage to enhance the quality of 3D instances and global layout. First, we develop collision-aware layout optimization to adjust the layouts of the objects to create a globally coherent and visually appealing composition. By incorporating collision awareness, we prevent objects from severe intersections, resulting in a realistic compositional layout. Second, we carry out instance-wise refinement to further improve the geometry and texture of individual objects, which also allows the customization of different instances as an added benefit.

We evaluate our method with a proposed validation set, extensive experiments, and ablations to showcase the ability of Layout-Your-3D in rapid and accurate 3D generation. In addition, we conduct comparisons with current state-of-the-art text-to-3D methods, demonstrating the potential of Layout-Your-3D in compositional 3D generation, as well as customized design and editing. The contributions of this work are threefold:

- We propose Layout-Your-3D, a practical approach for highly controllable compositional 3D generation, demonstrating superior performance compared to the baseline methods.
- We present a reconstruction-based initialization process that efficiently produces reasonable coarse 3D scenes. We further enhance the texture, geometry, and global layout of the 3D scenes using a carefully designed two-step disentangled refinement stage.

- We conduct comprehensive experiments to assess the effectiveness of our proposed method against state-of-the-art baselines. Our approach demonstrates superior qualitative and quantitative results while significantly reducing the time for generation (i.e., 12 minutes).

## 2 RELATED WORK

**Text-to-3D generation.** Significant advances have been made in this field thanks to the rapid development of foundational diffusion models (Rombach et al., 2022). The DreamFusion (Poole et al., 2022) model enables the generation of 3D objects using powerful 2D diffusion models by introducing the Score Distillation Sampling (SDS). Numerous works further improve text-to-3D generation performance by developing various SDS methods (Yu et al., 2023; Ma et al., 2024), leveraging advanced 3D representations such as DMTet (Chen et al., 2023) and 3D Gaussian Splatting (3DGS) (Chen et al., 2024c; Liang et al., 2024), or leveraging pre-trained normal and depth diffusion models (Qiu et al., 2024). For instance, ProlificDreamer (Wang et al., 2023) proposes a particle-based variational framework that promotes the generation of more diverse and less saturated 3D objects, while GaussianDreamer (Yi et al., 2024) accelerates optimization by leveraging 3DGS. Since 3DGS enables fast generation speed with various initialization methods and state-of-the-art generation performance, we are building on it for 3D generation from text prompts.

**Image-to-3D Reconstruction.** Instead of using inherently ambiguous text descriptions, one line of works reconstructs 3D objects from one or more input images. Numerous methods (Liu et al., 2024a; Long et al., 2024; Shi et al., 2024) finetune 2D diffusion models with 3D-aware data to generate highly consistent multi-view images. Subsequent approaches, such as LRM (Hong et al., 2024) and LGM (Tang et al., 2024a), further accelerate 3D reconstruction from single-view images through an efficient feed-forward process. As these reconstruction models are trained on synthetic, single-object-centric datasets, they often struggle to reconstruct plausible 3D assets that include multiple objects or produce realistic appearances for out-of-distribution input images. In this work, we leverage the efficient reconstruction models to produce initial coarse 3D scenes while addressing their limitations in compositional generation and the issue of unappealing appearances.

**Compositional 2D and 3D generation.** Generating or reconstructing scenes with multiple objects presents significant challenges in 2D and 3D contexts. In the 2D domain, numerous models (Li et al., 2023; Xie et al., 2023; Zhou et al., 2024a) use 2D bounding box layouts as conditions to guide the generation process. These works perform well in controlling the number of instances and their attributes. Due to the additional depth dimension, generating compositional assets in the 3D domain faces even more significant challenges. Several image-to-3D reconstruction methods, such as that by (Han et al., 2024; Chen et al., 2024b), reconstruct each object in the scene independently and subsequently optimize the global layouts by aligning them with reference images or utilizing additional guidance from diffusion models. Another line of work focuses on compositional text-to-3D generation. For example, COMOGen (Sun et al., 2024) adopts 2D layouts to guide the SDS process to provide accurate control over the generation process. GraphDreamer (Gao et al., 2024) establishes a graph to represent objects and their relations. GroundedDreamer (Li et al., 2024b) integrates the core idea of Attend-and-Excite (Chefer et al., 2023) into MVDream (Shi et al., 2024) to facilitate the generation of compositional multi-view images from complex text prompts. Moreover, text-to-3D scene generation methods (Zhou et al., 2024b; Li et al., 2024a; Po & Wetzstein, 2023; Zhang et al., 2024) that rely on the score distillation technique often utilize 3D layouts provided by users (Bai et al., 2024) or generated from LLMs to provide coarse compositional spatial information to construct the scene. However, these methods often require hours for generation and may produce unreasonable object interactions or unrealistic appearances. In this work, we leverage efficient reconstruction models to achieve high quality and efficiency, significantly advancing high-fidelity compositional 3D scene generation.

## 3 METHOD

Given a 2D layout and a text prompt, we aim to synthesize a 3D scene, including multiple objects and specific interactions aligned with the layout and text. Unlike existing text-to-3D methods, our approach aims to generate reasonable 3D geometry and texture for each object and produce a plausible layout with all objects interacting naturally (i.e., with reasonable location, scales, etc.).

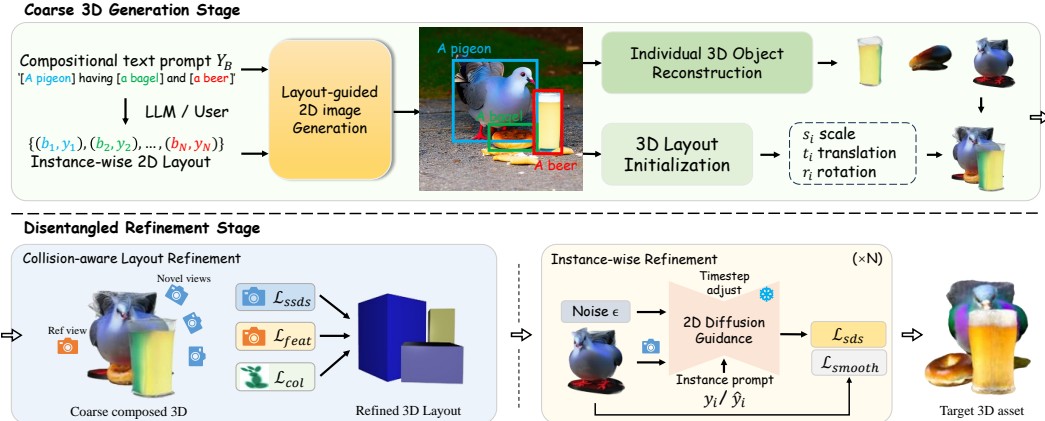

Figure 2: **Overview of Layout-Your-3D**. Given a 2D layout and text prompt, our coarse 3D generation stage (green box, see Sec. 3.1) generates coarse 3D instances along with roughly reasonable layouts. The disentangled refinement stage (see Sec. 3.2) then refines the 3D layout and enhances individual instance quality by leveraging a collision-aware layout refinement (blue box) followed by an instance-wise refinement (yellow box).

As shown in Fig. 2, we represent the 2D layout as a set of bounding boxes and their corresponding instance names $B = \{(b_1, y_1), (b_2, y_2), ..., (b_N, y_N)\}$, which can be extracted from the overall text prompt $Y_B$. Given these bounding boxes and instance names, we design a coarse 3D generation stage (see Sec. 3.1) that creates all 3D objects along with a roughly reasonable 3D layout. To enhance the visual quality and consistency of the layout, we introduce a disentangled refinement stage (see Sec. 3.2), which includes a collision-aware layout refinement, followed by an instance-wise refinement. Additionally, our method allows users to interactively edit each instance and the global layout as a by-product. In the following, we provide detailed discussions of these components.

## 3.1 COARSE 3D GENERATION STAGE

Given a set of bounding boxes with corresponding object names, the coarse 3D generation stage synthesizes each 3D object and arranges them in a roughly reasonable layout. To achieve this, we begin by creating a reference image based on the 2D layout. Next, we reconstruct each 3D object (instance) using an efficient reconstruction model. We adopt 3DGS as our 3D representation, considering its flexibility and geometry attributes. For layout initialization, naively converting the 2D layout into 3D space using depth information can lead to unrealistic overlaps and collisions between objects. To address this issue, we develop an initialization process to produce plausible 3D layouts by incorporating both geometry and semantic knowledge.

**Reference Image Generation.** Instead of lifting 2D bounding boxes to 3D space and generating 3D objects from scratch (Sun et al., 2024), our method first synthesizes a 2D reference image based on the given layout and text prompt. This reference image significantly narrows the solution space and provides crucial guidance for the subsequent generation process. Specifically, we use the state-of-the-art method MIGC (Zhou et al., 2024a) for layout-guided 2D image generation, defined as:

$$I_{ref} = \texttt{MIGC}(Y_B, B), \tag{1}$$

where $I_{ref}$ represents the generated reference image.

**Individual 3D Object Reconstruction.** Next, we reconstruct each instance from the generated 2D reference image. To alleviate the influence of occlusion among instances, we first segment each instance in the reference image $I_{ref}$ using Segment-Anything Model (SAM) (Kirillov et al., 2023), and inpaint them as follows:

$$I_i, M_i = \texttt{SAM}(I_{ref}, b_i); \quad \hat{I}_i = \texttt{SD}(I_i, (\sim M_i) \cap b_i, y_i), \tag{2}$$

where $I_i$ and $M_i$ are the RGB values and the foreground mask of the $i_{th}$ instance. $SD$ is the Stable Diffusion inpainting model, $(\sim M_i) \cap b_i$ represents the region to be inpainted and $\hat{I}_i$ is the $i_{th}$ completed 2D instance. Note that to make the inpainted image compatible with the object-centric reconstruction models, we also perform post-processing to remove the background and center

the object within the image $\hat{I}_i$. Fig. 3 illustrates this segmentation, inpainting and post-processing process. Since the reconstruction models are sensitive to input image quality, we optionally enhance the details of the inpainted image $\hat{I}_i$ using a ControlNet-Tile (Zhang et al., 2023) model. Finally, $\hat{I}_i$ is fed into the Large Multi-View Gaussian Model (LGM) (Tang et al., 2024a) to generate the coarse 3D instance $A_i$. Thanks to the explicit geometry priors from LGM and attributes of gaussian splatting, we could primarily eliminate the Janus Problem and add fine-grained details to the original instances in the subsequent refinement step.

**3D layout initialization.** Given the synthesized 3D instances, we arrange them in the 3D space by lifting the 2D bounding boxes $\{b_i\}$ discussed above into 3D. Each 3D bounding box is defined by its scale, rotation, and translation (denoted as $\{s_i, r_i, t_i\}$). From the given 2D layout, the scale $s_i$ of each 3D box can be intuitively set to $W_{b_i}/\overline{W_{b_i}}$, in which $W_{b_i}$ and $\overline{W_{b_i}}$ are the width of original and post-processed 2D box $b_i$ (see Fig. 3). For the translation $t_i$, we can directly get the first two elements (i.e., $x_i, y_i$) of the transition vector $t_i = \{x_i, y_i, z_i\}$ as the center coordinates

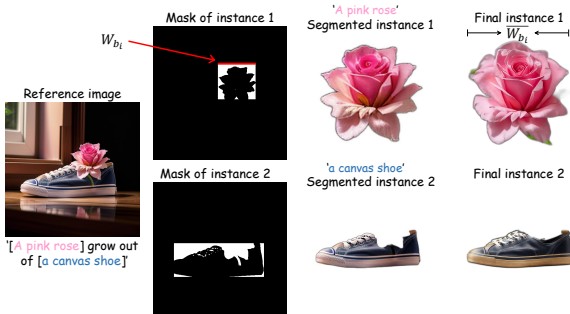

Figure 3: Example of the instance segmentation, inpainting and post-process.

of $b_i$. Additionally, $z_i$ is initialized as the average of the foreground depth value predicted by an off-the-shelf depth estimation model (i.e., GeoWizard (Fu et al., 2024)), which is computed as:

$$z_i = \text{mean}\left(M_i \odot \text{Depth}(I_{ref})\right). \tag{3}$$

Rotation of each instance is crucial in accurate layout estimation. Unlike existing works that overlook rotation initialization, we propose to estimate rotation based on feature similarity between rendered images and the 2D instance $\hat{I}_i$ in the reference image. Specifically, we render the coarse 3D instance $A_i$ from a set of $n$ camera poses with uniformly sampled azimuth and elevation, denoted as $\pi \in \mathbb{R}^{n \times 2} = \{(e_j, a_j)\}_{j=1}^{n}$. We then extract the features of 2D instance in the reference images and rendered images as $f_{\hat{I}_i}$ and $F = \{f_1, f_2, ..., f_n\}$ from DINOv2 (Oquab et al., 2023), respectively. Finally, we calculate the cosine similarity and choose the best rotation $r_i$:

$$(e_b, a_b) = \arg\max_{(e,a)}(cos(f_{\hat{I}_i}, F)), \tag{4}$$

where $(e_b, a_b)$ is elevation and azimuth of the best rotation $r_i$. As a result, our reconstruction-based initialization strategy provides a more accurate initial state for the final compositional 3D, which can benefit the subsequent refinement stage.

## 3.2 DISENTANGLED REFINEMENT STAGE

Previous works (Zhou et al., 2024b; Li et al., 2024a) have attempted to optimize 3D scenes through local and global guidance. However, they often suffer from excessively long optimization times and the need for additional guidance. Instead, we disentangle the refinement process into two consecutive steps: a collision-aware layout refinement step and an instance-wise refinement step.

### 3.2.1 COLLISION-AWARE LAYOUT REFINEMENT

The Coarse 3D Generation Stage (Sec. 3.1) has provided a relatively precise initial 3D layout, however, we empirically observe that it is still insufficient to create a visually appealing composition. Thus we propose a layout refinement step for further improvement. First, we adopt the spatial-aware SDS Loss (SSDS) as proposed in (Chen et al., 2024b):

$$\nabla_\theta \mathcal{L}_{ssds}(\phi^*, x) = \mathbb{E}_{t,\epsilon}[w(t)(\hat{\epsilon}_{\phi^*}(x_t; Y_B, t) - \epsilon)\frac{\partial x}{\partial \theta}], \tag{5}$$

In which $\theta, x, \phi^*, \hat{\epsilon}_{\phi^*}(x_t; Y_B, t)$ are the 3D representation, rendered image, attention map strengthened on spatial words, and the score function that predicts the sampled noise $\epsilon$ from the noised

image $x_t$ with noise-level $t$ and text prompt $Y_B$. However, the SSDS loss is inadequate for addressing significant collisions caused by occlusion. For example, in Fig. 8, the dog obscures the blue tie and consequently does not receive sufficient gradients during optimization. To resolve this issue, we propose a feature-level reference loss that leverages the 2D reference image and a tolerant collision loss to handle object collisions more effectively.

**Feature-level Reference Loss.** To align the geometry information in the reference image with our 3D scene, the most intuitive method is to apply a pixel-level RGB loss to enforce the alignment. Nevertheless, we notice that the post-processing and the reconstruction model will unavoidably introduce some distortions and artifacts, making the RGB loss less robust and functional. Therefore, we propose a feature-level reference loss to stabilize the refinement process. Specifically, we only keep the foreground instances in the reference image and obtain its feature from higher layers of the DINOv2 (Oquab et al., 2023) model:

$$f_{ref} = \texttt{DINO}(I_{ref} \odot (M_1 \cup M_2 \cup ... \cup M_N)). \tag{6}$$

The feature-level reference loss can be computed as the L2 distance of $f_{ref}$ and $f_{render}$, here $f_{render}$ is the extracted feature of the rendered image at the reference view:

$$\mathcal{L}_{feat} = \lambda_f \sum \|f_{ref} - f_{render}\|_2 . \tag{7}$$

**Tolerant Collision Loss.** Inspired by physics-based simulation (Santesteban et al., 2021) and CG3D (Vilesov et al., 2023), we propose a tolerant collision loss to separate incorrectly overlapping instances while simultaneously allowing a certain degree of collision to model natural object interactions. As illustrated in Fig. 4, gaussians for two interacting instances are denoted as $P_1 = \{p_1^1, p_2^1, ..., p_{K_1}^1\}$ and $P_2 = \{p_1^2, p_2^2, ..., p_{K_2}^2\}$, where $p_k$ is the coordinate of the $k_{th}$ gaussian. The $K_1$ and $K_2$ are the numbers of gaussians in the two instances, respectively. In order to get a rough estimation of each instance, we first compute the mean coordinate of each instance de-

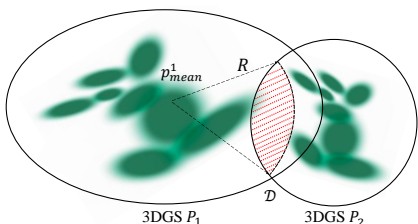

Figure 4: A simple illustration of how to calculate the collision loss $\mathcal{L}_{col}$.

noted as $p_{mean}^1$ and $p_{mean}^2$. Then, we define mean sparsity $R$ as the mean distance of each point in $P_1$ to $p_{mean}^1$, which can also be interpreted as calculating the mass distribution of each instance:

$$R = \frac{1}{K} \sum_{k=1}^{K} \|p_k^1 - p_{mean}^1\|_2 . \tag{8}$$

Next, we calculate distances from all the points in $P_2$ to $p_{mean}^1$ and subtract mean sparsity $R$, which can be regarded as the average distance between the distributions of two instances. The tolerant collision loss $\mathcal{L}_{col}$ is computed as:

$$\mathcal{L}_{col} = \lambda_c \sum_{k=1}^{K_2} ReLU(\mathcal{D}) \quad \text{where } \mathcal{D} = \{R - \|p_k^2 - p_{mean}^1\|_2, k = 1, 2, ..., K_2\}. \tag{9}$$

Here $ReLU$ is the rectified linear unit. Intuitively, this loss penalizes gaussians that are too close to the centers of other instances, while allowing specific distances to model intersection between instances, making the optimization more robust and effectively addressing the overlapping problem.

Our full collision-aware layout loss is defined as a weighted combination of the three loss terms:

$$\mathcal{L}_{layout} = \mathcal{L}_{ssds} + \lambda_f \mathcal{L}_{feat} + \lambda_c \mathcal{L}_{col}. \tag{10}$$

### 3.2.2 INSTANCE-WISE REFINEMENT

Since the LGM can only produce fixed number of 3D gaussians, the reconstructed 3D instances are often in low quality with holes and broken textures. Next, we discuss refining these coarse 3D instances to enhance its geometry and texture quality. The conventional refinement strategies adopted by other optimization-based methods (Wang et al., 2023; Yi et al., 2024) are not efficient enough and often result in complete deformations of the shape of coarse 3D instances due to the

excessive noise added to the rendered view, which aggravates the degeneration of geometry and results in drastic changes. To address this issue, we propose an adjustable timestep sampling strategy to preserve the geometry details and retain the primary interactions of the coarse 3D instances. Specifically, we lower the upper bound of timestep sampling's range in the earlier phase and increase it after specific iterations when the geometry details have been well formed. Formally, the gradient of instance refinement SDS loss $\hat{\mathcal{L}}_{sds}$ is computed as follows:

$$\nabla_\theta \hat{\mathcal{L}}_{sds}(\phi, x) = \mathbb{E}_{\hat{t}, \epsilon}[w(\hat{t})(\hat{\epsilon}_\phi(x_{\hat{t}}; y_i, \hat{t}) - \epsilon)\frac{\partial x}{\partial \theta}], \qquad (11)$$

where $\hat{t}$ is the adjusted timestep. Intuitively, geometry is most likely to deviate during earlier refinement phase but should remain mostly unchanged in the rest of the process. As a result, our minor modification prevents the basic geometry from deviating significantly from its initial state. Moreover, we noticed that the normal of coarse 3D instances might be rough and contain artifacts, which can harm their visual quality. So we combine an additional normal smooth loss $\mathcal{L}_{smooth}$ and total variation (Rudin & Osher, 1994) (TV) regularization terms (denoted as $\mathcal{L}_{TV}^d$ and $\mathcal{L}_{TV}^n$) to encourage better geometry of 3D instances. The full loss for instance-wise refinement can be represented as:

$$\mathcal{L}_{instance} = \hat{\mathcal{L}}_{sds} + \lambda_s \mathcal{L}_{smooth} + \lambda_{tv}(\mathcal{L}_{TV}^d + \mathcal{L}_{TV}^n). \qquad (12)$$

It is worth noting that when we extend the iterations for refinement (marked as extended refinement), we can produce comparable or even better single-object generation results compared to other optimization-based methods (Liang et al., 2024; Wang et al., 2023), as shown in Fig. 6.

**Interactive Editing.** Besides 3D scene generation, users can customize the per-instance text prompt $\hat{y}_i$ in the instance-wise refinement step for interactive editing. Specifically, we first generate a 3D scene with our coarse 3D generation stage and then stylize each 3D instance in the refinement step, with users' custom text prompts $\hat{y}_i$. This editing application opens up new possibilities for generating extra-complex compositional 3D scenes, with each instance having its attribute and style.

## 4 EXPERIMENTS

### 4.1 IMPLEMENTATION DETAILS

In the Coarse 3D Generation Stage, we render the 3D instances at 10-degree intervals when estimating the coarse rotation $r_i$. For the Disentangled Refinement Stage, in the collision-aware layout refinement step, we use the original hyper-parameter settings for the SSDS loss $\mathcal{L}ssds$ and optimize the layout for 400 iterations, with $\lambda_f$ and $\lambda_c$ set to 10.0 and 0.2, respectively. In the instance-wise refinement step, we utilize DeepFloyd (Shonenko et al., 2023) guidance with a total of 1500 iterations following the short refinement strategy (see Fig. 5). We set the timestep range to [0.10, 0.50] from step 0 to 800, and [0.02, 0.75] from step 800 to 1500. The parameters $\lambda_s$ and $\lambda_{tv}$ are set to 1.0 and 0.2, respectively. The extended instance-wise refinement strategy and additional implementation details are provided in Appendix B.

### 4.2 MAIN RESULTS

**Validation Set.** To better validate Layout-Your-3D's ability in compositional 3D generation, we construct a validation set naming *Compo20* for evaluation. The *Compo20* consists of 20 compositional text prompts, each containing two or more instances with specific interactions. For each text prompt, we have a user to manually provide a 2D layout and also automatically generate a 2D layout using LLM-grounded Diffusion (Lian et al., 2023).

**Comparison methods.** We compare Layout-Your-3D with several text-to-3D methods, i.e., DreamFusion (Poole et al., 2022), ProlificDreamer (Wang et al., 2023), GaussianDreamer (Yi et al., 2024), LucidDreamer (Liang et al., 2024) and GraphDreamer (Gao et al., 2024). Since our method utilizes an additional 2D layout as a condition, we provide results by both the user-given and LLM-grounded 2D layout. Moreover, we present a comparison of single 3D object generation in Fig. 6. Note that we do not compare with works that haven't released complete or official implementations (e.g., Locally Conditioned Diffusion (Po & Wetzstein, 2023), CG3D (Vilesov et al., 2023)).

**Qualitative Comparison.** As shown in Fig. 5, our method can generate compositional 3D scenes with higher quality and better understand the text prompts. In contrast, other methods often struggle

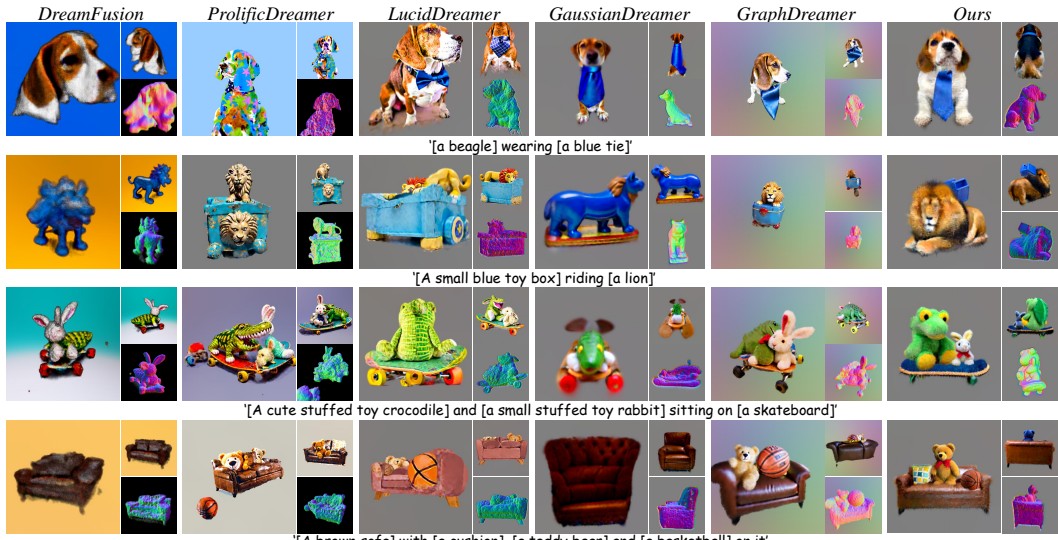

*DreamFusion*    *ProlificDreamer*    *LucidDreamer*    *GaussianDreamer*    *GraphDreamer*    *Ours*

'[a beagle] wearing [a blue tie]'

'[A small blue toy box] riding [a lion]'

'[A cute stuffed toy crocodile] and [a small stuffed toy rabbit] sitting on [a skateboard]'

'[A brown sofa] with [a cushion], [a teddy bear] and [a basketball] on it'

Figure 5: Main results and comparison with other text-to-3D methods on our *Compo20* validation set. Layout-Your-3D can generate compositional 3D scenes with higher quality and more reasonable 3D layouts. Note that the first two rows of our results are generated with LLM-grounded 2D layouts, and the last two are generated with user-given 2D layouts.

Table 1: Quantitative results of Layout-Your-3D and other text-to-3d methods on our proposed *Compo20* validation set. Note that we adopt the short instance-wise refinement setting here for fair comparison. The best and second best results are **bold** and underlined, respectively.

| Method | Time cost ↓ | CLIP-Score ↑ | BLIP-VQA Score ↑ | mGPT-CoT ↑ |
|---|---|---|---|---|
| DreamFusion | 45 mins | 17.98% | 23.44% | 31.08% |
| ProlificDreamer | 4 hours | **24.93%** | 42.14% | 21.36% |
| GaussianDreamer | 15 mins | 22.21% | 28.74% | 25.91% |
| LucidDreamer | 35 mins | 22.36% | 34.11% | 34.84% |
| GraphDreamer | 2 hours | 24.78% | 31.18% | 44.28% |
| Layout-Your-3D | 12 mins | 23.26% | **53.51%** | **50.44%** |
| Layout-Your-3D (LLM-grounded) | | 22.83% | 52.26% | 48.80% |

to comprehend spatial relationships and semantic attributes in text prompts, leading to poor quality and the Janus Problem. These results highlight Layout-Your-3D's strength in compositional 3D generation. More visualization results are provided in the appendix C.

Table 2: User study on different text-to-3D methods in terms of text-alignment, quality and rationality. The best and second best results are shown in **bold** and underlined.

| Method | Text-Alignment | Quality | Rationality |
|---|---|---|---|
| DreamFusion | 3.36 | 5.48 | 3.79 |
| ProlificDreamer | 7.34 | 7.26 | 6.91 |
| GaussianDreamer | 4.22 | 6.61 | 4.72 |
| LucidDreamer | 5.61 | 7.72 | 6.38 |
| Ours | **9.08** | **8.44** | **8.72** |
| Ours(LLM-grounded) | 8.68 | 7.93 | 8.14 |

**Quantitative Comparison.** We conduct quantitative comparisons on the *Compo*20 validation set, presenting results adopted by the short refinement strategy. Since CLIP-Score (Radford et al., 2021) is not capable of accurately measuring the semantic correspondences (Huang et al., 2023a), we also employ other metrics, BLIP-VQA and mGPT-CoT (Huang et al., 2023a), to make a fine-grained assessment of the text-3D alignment. As shown in Tab. 1, our Layout-Your-3D outperforms other methods, demonstrating the superiority of our method on quantitative evaluation.

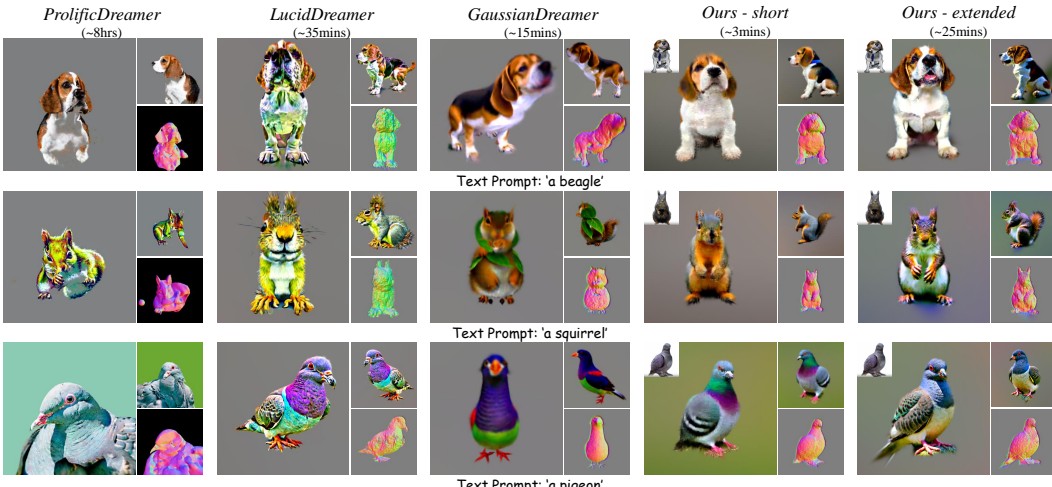

Figure 6: Comparison on the quality of single 3D instance generation. We provide both the short and extended strategy. When lengthening the optimization process, our Layout-Your-3D can generate comparable or even better results than SOTA text-to-3D methods.

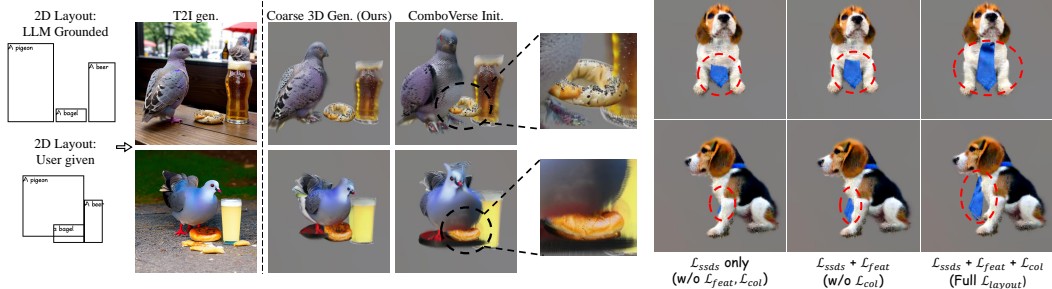

Figure 7: Illustration of user given and LLM-grounded 2D layout generated reference images. We also visualized our coarse 3D generation and compared it with ComboVerse.

Figure 8: Ablation study on the effectiveness of different loss items in our layout loss $\mathcal{L}_{layout}$.

**User Study.** In addition to qualitative and quantitative comparisons, we conduct a user study to evaluate the proposed method further. Participants are asked to rate text alignment, quality, and rationality on a scale ranging from 1 to 10. Concretely, users are shown 3D results and corresponding text prompts in a shuffled order and then asked to give ratings. We collect 500 responses from 100 users. In Tab. 2, the overall ratings indicate a clear preference of users for our method.

## 4.3 ABLATION STUDIES

**3D layout initialization.** We conduct an ablation study on the 3D layout initialization strategy by comparing the generated coarse scenes. As illustrated in Fig. 7, compared to ComboVerse (Chen et al., 2024b), our initialization provides a more accurate starting point for subsequent refinement. These results demonstrate the effectiveness of our initialization strategy.

**Collision-aware Layout Refinement.** We analyze the function of various loss terms in the layout refinement step by using a toy example. When different 3D instances have major intersections, the overlapped instances tend to be concealed in the rendered images, thus hindering the optimization of diffusion-guided loss. Instead, our tolerant collision loss $\mathcal{L}_{col}$ further optimizes the geometry itself and $\mathcal{L}_{feat}$ encourages the alignment of spatial and high-level semantic information between the reference image and rendered reference view. As shown in Fig. 8 and Tab. 3, our layout loss $\mathcal{L}_{layout}$ can effectively address this issue, thus resulting in more coherent 3D layouts.

**Instance-wise Refinement.** We conduct an ablation study on the influence of our timestep adjustment given examples with different settings. As shown on the left side of Fig. 9, timestep adjustment strategies applied by other methods often result in inferior performances and would introduce artifacts in the appearance. On the right side of Fig. 9, we visualize the effects of the

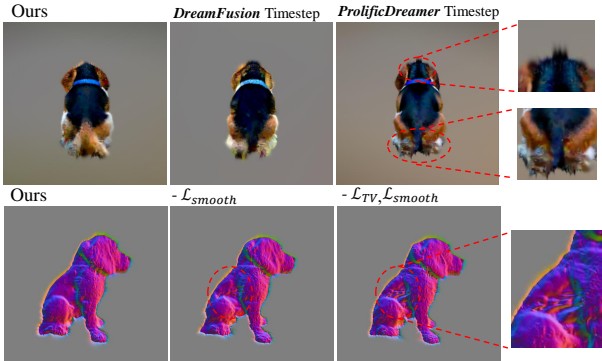

Figure 9: Ablation studies on timestep sampling and additional loss items in instance-wise refinement step.

Table 3: Ablation study on different loss items of our $\mathcal{L}_{layout}$, measured in BLIP-VQA score.

| Loss term | BLIP-VQA ↑ |
|---|---|
| Full $(\mathcal{L}_{layout})$ | 53.51% |
| $-\mathcal{L}_{col}$ | 52.04% |
| $-\mathcal{L}_{ssds}$ | 52.71% |
| $-\mathcal{L}_{feat}$ | 53.41% |
| $-\mathcal{L}_{feat}, \mathcal{L}_{col}$ | 53.09% |

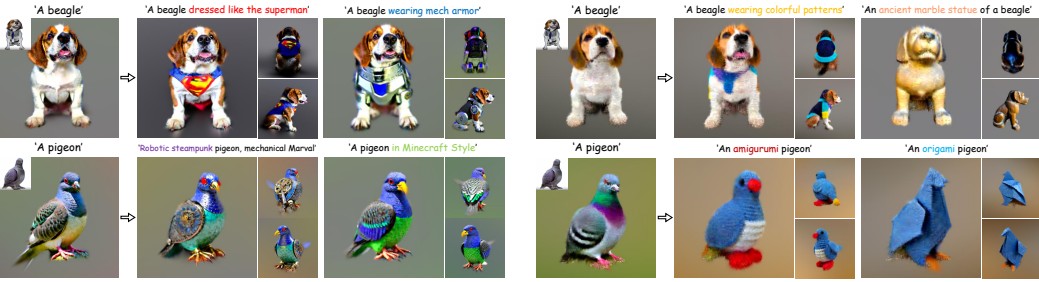

(a) The extended refinement strategy.  (b) The short refinement strategy.

Figure 10: Examples on the 3D instances refined with custom text prompts. We conduct experiments on both the longer and shorter refinement strategies to validate the effectiveness of customization.

TV losses $\mathcal{L}_{TV}^d, \mathcal{L}_{TV}^n$ and normal smooth loss $\mathcal{L}_{smooth}$ By incorporating these two additional loss terms, Layout-Your-3D can achieve better geometry consistency and quality. We note that all the analyses above are based on a short instance-wise refinement strategy.

### 4.4 DOWNSTREAM APPLICATIONS

**Instance customization.** We present 3D instances refined with custom text prompts in Fig. 10. Our method can guide the refinement process to generate plausible and coherent results that are more aligned with text prompts. Thus, Layout-Your-3D allows for the initial generation of a coarse 3D scene, followed by the controlled adjustment of specific attributes across different instances. As a result, our method can produce stylized and complex outputs with better flexibility.

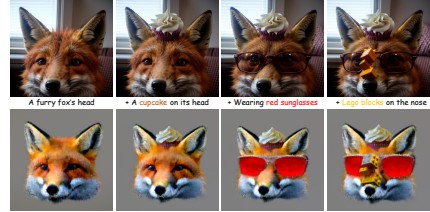

Figure 11: A simple illustration of the object insertion application.

**Object insertion.** Since 3D results are closely related to the reference image, incorporating object insertion into our pipeline would be straightforward. As illustrated in Fig. 11, the first and second rows show the edited reference images and corresponding 3D results. We can see that our method can insert objects with iterative editing, helping users customize their desired 3D content.

## 5 CONCLUSION

In this paper, we propose Layout-Your-3D, a method designed for highly controllable and precise compositional 3D generation. Our method addresses the issue of compositional 3D generation in a coarse-to-fine manner. First, we initialize the rough asset with our Coarse 3D Generation Stage. Then, we propose a Disentangled Refinement Stage. By refining the layout and instances separately, Layout-Your-3D can achieve more flexible and high-quality compositional 3D generation with a minimal cost of time. Through massive experiments and comparisons, we believe Layout-Your-3D indicates an essential step toward more comprehensive and mature 3D generation.

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

## A  DISCUSSIONS

### A.1  LIMITATIONS & FUTURE WORK

Since Layout-Your-3D relies on reference images to generate compositional 3D assets, it would need deliberately designed 2D layouts to generate extra-complex 3D scene, e.g., having more than

10 instances in the asset. What's more, as we observed, the SSDS loss, proposed in ComboVerse, is not capable of harmonizing interactions with too many instances involved.

Thus, in our future work, we will work on deliberately designed modules to achieve more mature text-to-3d layout refinement and generation, without constraints on the number of instances or interactions. Also, the application in scene generation of our method is a promising direction, we can extend our method in order to improve the efficiency and controllability of Text-to-3D scene generation.

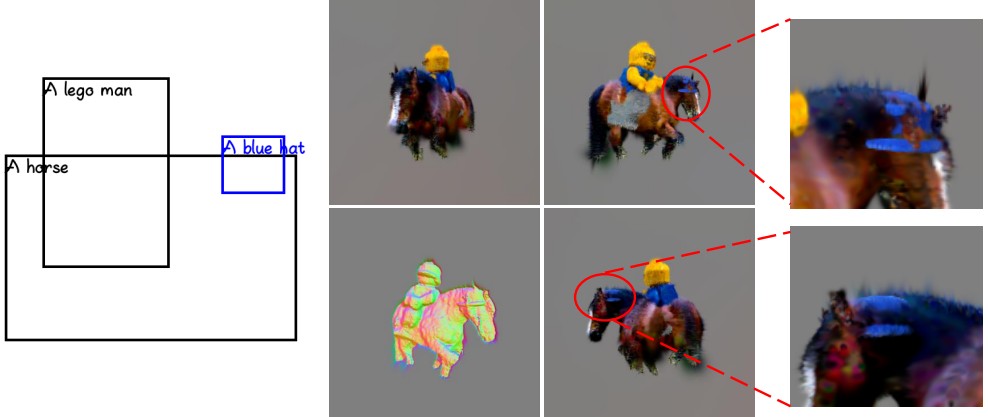

Figure 12: A simple illustration of failure cases.

## A.2 FAILURE CASES

We provide a simple failure case in Fig 12. The text prompt is *'A lego man riding a horse, the horse is wearing a blue hat'*, in which we add uncommon interactions between instances, i.e., *'blue hat'* and *'horse'*. As illustrated in the figure, for text prompts that are extremely complicated or counter-intuitive, the diffusion models are not capable of comprehending the spatial relationships correctly and hence misdirect the optimization process.

## A.3 MORE DOWNSTREAM APPLICATIONS

Except for instance-wise customization and object insertion, our Layout-Your-3D can also support the Conversational Iterative Editing as proposed in GALA3D. Once obtained the final 3D layouts of the resulting compositional 3D, we can easily edit the position and rotation of each instance, together with its attributes and style, through iterative conversation. Since the generated 3D asset is essentially comprised of different 3D instances, it has a higher degree of freedom and can achieve better quality. This advantage helps Layout-Your-3D stand out among different Text-to-3D generation methods.

## A.4 DISCUSSION ON CONCURRENT METHODS

GroundedDreamer (Li et al., 2024b) and COMOGen (Sun et al., 2024) are the two most related and concurrent works. Here we give a comprehensive analysis and a comparison between these two methods and Layout-Your-3D. Since they have not released official implementations yet, we simply use the results from the original paper for comparison, and analyze the possible advantages and disadvantages.

**GroundedDreamer**'s core concept is to marry Attend-and-Excite with MVDream (Shi et al., 2024) to generate compositional and consistent multi-view images, then lift them to 3D assets through an optimization-based method. Though it shows some surprising results and relatively high quality, due to the limited compositional 3D data used in the training process, the limitations of MVDream still exist, i.e., lack of prior knowledge for generating multiple instances. Also, the pure optimization-based paradigm makes GroundedDreamer low in efficiency, taking nearly 2 hours to generate one single asset. In comparison, Layout-Your-3D needs only about ten minutes (short refinement) or

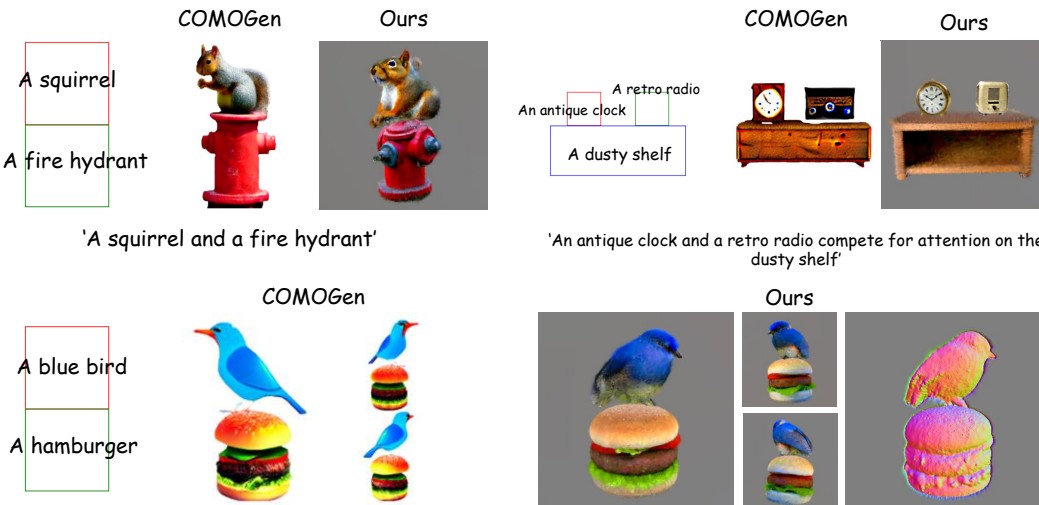

Figure 13: Comparison with COMOGen, results are generated with 2D layouts provided in the original paper of COMOGen.

about one hour (extended refinement) to generate one asset. If given more GPU cards, the time cost could be reduced to several minutes and half an hour. Furthermore, since Layout-Your-3D primarily generates individual 3D instances and subsequently combines them, which mitigates the domain gap during the generation process, it effectively helps address compositional 3D generation task at its source.

**COMOGen** takes the same 2D layout setting as adopted in Layout-Your-3D to indicate ideal regions of different instances in the text prompts. However, different from Layout-Your-3D, COMOGen first adopted a Layout-SDS loss to distill the spatial layout knowledge, then uses a Multi-view Control module to ensure multi-view consistency and alleviate Janus Problem, finally proposed a 3D content enhancement module to generate realistic and diverse 3D samples. The Layout-SDS loss could provide a rough estimation of the ideal 3D content, in the following Multi-view Control module, COMOGen adopted a Zero-123 (Liu et al., 2023) multi-view diffusion model to ensure the geometry consistency. But as we mentioned above, current multi-view diffusion models are still subject to the domain gap, thus making it hard to generalize to compositional scenarios, damaging the effectiveness of COMOGen. Moreover, though not presented, the time cost for COMOGen might not be comparable to our Layout-Your-3D, which only costs about 10 minutes to generate a reasonable and plausible compositional 3D scene, as displayed in Fig. 13. Additionally, COMOGen did not show detailed and high-quality single object generation results, which constrains the application scenarios of it.

Another line of works, i.e., **GALA3D** and **DreamScene** also have similar settings as our Layout-Your-3D. Nonetheless, these works have their own disadvantages. DreamScene is essentially composing different objects into one predefined scene with no specific interaction modeling process during generation, thus making the resulting scene less realistic and consistent. GALA3D addresses this issue by incorporating 3D layout-conditioned diffusion models as an implicit control over the interactions between objects. However, the produced 3D scene has no background and the quality of objects is relatively inferior, though processed through a tedious optimization process. Since the input settings and generated 3D assets are not strictly aligned with ours, we do not make direct comparisons between these methods and our Layout-Your-3D.

## A.5 OTHER RECONSTRUCTION MODELS

In the coarse 3D generation stage, we could exchange for any 3DGS-based Image-to-3D model for the reconstruction of each instance, e.g., GRM (Xu et al., 2024), Point-to-Gaussian (Lu et al., 2024). What's more, it is also applicable to replace 3DGS with other 3D representations, like mesh or tetrahedron for advanced or specific usage. Though it will be slightly different to optimize coarse

3D instances in other representations, there are substitutions for the refinement strategies in our pipeline. Also, our final 3DGS-based asset could be converted into mesh with the efficient mesh extraction method proposed in DreamGaussian (Tang et al., 2024b) or advanced mesh extraction methods like SuGaR (Guédon & Lepetit, 2024).

# B    EXPERIMENT SETTINGS

## B.1    VALIDATION SET

We analyze the core design of our proposed $Comp20$ validation set in this part. In order to provide a comprehensive measurement for the compositional 3D generation task, we divide the text prompts in our $Comp20$ validation set into several scenarios, covering a wide range of cases that can meet the requirements by users and real world applications:

- Multiple instance without interactions,
  e.g., *'A black bottle, with a white bottle on one side and a blue bottle on the other'*.

- Multiple instances with interactions,
  e.g., *'An astronaut sitting on a sofa'*.

- Attribute control of instances,
  e.g., *'A blue toy robot and a red toy robot playing basketball'*.

- Small components in a larger main body,
  e.g., *'A furry fox's head, with a cupcake on its top, wearing red sunglasses and have a lego block on the nose'*.

In accordance with the requirements for compositional 3D generation, each text prompt contains 2 to 5 instances in the compositional 3D asset. This intentional design of our $Comp20$ validation set allows for a comprehensive evaluation on text-to-3D generation methods. The complete prompt list, together with the corresponding 2D layouts (including both user-provided and LLM-generated layouts), are presented in Tab. 9. Note that in the default experiments we generate reference images with a 512 * 512 resolution, and the 2D layouts are presented in the form of $[x_1, y_1, x_2, y_2]$, with each element has a value from range 0 to 512.

## B.2    MORE IMPLEMENTATION DETAILS

We give a more detailed illustration of our experiment settings in this section. We implement Layout-Your-3D based on threestudio (Guo et al., 2023) for a more integrated and readable system. We conduct all of our experiments based on NVIDIA RTX6000 GPUs. It is worth noting that the baseline methods, i.e., DreamFusion and ProlificDreamer are implemented on threestudio, thus might result in minor differences with the original implementation results.

Table 4: The detailed list of the short (efficient) instance-wise refinement step.

| Hyper-parameters | value |
|---|---|
| batch_size | 1 |
| resolution | [256, 512] |
| resolution milestones | 800 |
| position_lr | [0, 0.0005, 0.00005, 500] |
| scale_lr | 0.005 |
| feature_lr | 0.01 |
| opacity_lr | 0.01 |
| rotation_lr | 0.001 |
| densify/prune interval | 100 |
| densify/prune start iter | 300 |
| densify/prune until iter | 900 |
| total_iter | 1500 |

Table 5: The detailed list of the extended (high quality) instance-wise refinement step.

| Hyper-parameters | value |
|---|---|
| batch_size | 4 |
| resolution | 512 |
| position_lr | [0, 0.0005, 0.00002, 1000] |
| scale_lr | 0.005 |
| feature_lr | [0, 0.01, 0.005, 2000] |
| opacity_lr | 0.05 |
| rotation_lr | 0.005 |
| densify/prune interval | 200 |
| densify/prune start iter | 400 |
| densify/prune until iter | 1600 |
| total_iter | 2000 |

Table 6: The detailed hyper-parameter list of the collision-aware layout refinement step.

| Hyper-parameters | value |
|---|---|
| quaternion_lr | 0.0001 |
| transition_x_lr | 0.00002 |
| transition_y_lr | 0.00002 |
| transition_z_lr | 0.02 |
| total_iter | 400 |

In the coarse 3D generation stage, to guide the layout-conditioned text-to-image generation to generate canonical views, we add an additional text prompt *'3d scene, front view'* to the overall text prompt $Y_B$. This simple measure helps Image-to-3D reconstruction models obtain better results in the coarse 3D generation stage since multi-view diffusion models, e.g., MVDream, perform better when given images with canonical views. When performing segment & inpaint, we prompt the SAM (Kirillov et al., 2023) with 2D bounding box of each instance to indicate the foreground region and the center points of all other instance's boxes are treated as negative prompts to avoid segmenting multiple irrelevant instances all at once. Moreover, for the depth estimation part, since the output prediction from GeoWizard is not metric depth, we normalize the predicted depth values and centralize the 3D location of the final 3D assets. In the instance-wise refinement step, we provide a more concrete setting for both the extended and short refinement and layout refinement steps. All the hyper-parameters are listed in Tab. 4, Tab. 5, and Tab. 6, including learning rates for attributes of the gaussian splatting representation and different elements in layouts. Note that for the extended instance-wise refinement strategy we use Stable Diffusion (Rombach et al., 2022) guidance instead of Deep Floyd, to boost the quality of each instance. For the layout learning part, we assume that the input 2D layouts are accurate and reasonable enough. Since two of the spatial coordinates in the transition $t_i$ depend mainly on the given 2D layouts, we only give a very small learning rate to these two elements.

## B.3 ADDITIONAL ABLATION STUDIES

We conduct more ablation studies to validate the effectiveness of our model selection. First, we compare the visual foundation model used when extracting features for coarse rotation estimation. We choose DINOv2 (Oquab et al., 2023) to estimate the rotation $r_i$ since DINOv2 can provide higher-level geometry and semantic information. In comparison, CLIP (Radford et al., 2021) is another model we adopted, but since it is not specifically designed to capture geometry information in images, it causes visible errors in the rotation estimation.

Also, we give a detailed time cost of Layout-Your-3D in Tab. 7, including the time cost for each part and different settings, measured on one single RTX6000 GPU.

Moreover, to prove that it is essential to process the layout refinement step and instance-wise refinement step, we present some toy results after the Coarse 3D Generation stage in Fig. 14. These

Table 7: Time cost for each part of Layout-Your-3D, including both the short and extended refinement strategy.

| Designed module | Time cost |
|---|---|
| Coarse 3D gen. | + ~1 minute |
| Layout ref. | + ~2 minutes |
| Ins. ref. (short/extended) | + ~3 / 35 minutes |
| Overall (short) | ~12 minutes (3 instances) |

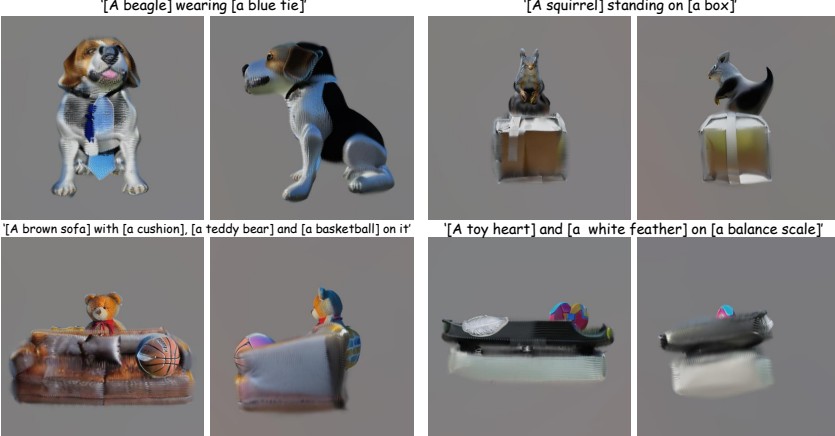

Figure 14: 3D assets generated without coarse 3D generation stage.

examples, though show relatively reasonable geometry shapes and spatial relationships, are still not satisfying enough for further usage. Thus we claim that it is necessary for us to develop the Disentangled Refinement Stage to obtain better results.

## C MORE QUALITATIVE RESULTS

### C.1 SINGLE OBJECT GENERATION

In Fig. 15, we visualize more single object generation results produced by our extended instance-wise refinement step. These results demonstrate the capability of our Layout-Your-3D in the field of text-to-single 3D object generation. The figure shows that our refinement process could retain the primary geometry shapes and add fine-grained details. In about 25 to 30 minutes, we can generate 3D object of extra high quality with our Layout-Your-3D, comparable, or even much better than previous or current SOTA methods. These results proves the versatility of our method.

### C.2 CUSTOMIZATION

To validate the effectiveness and generalizability of our method, we give more examples in Fig. 16, both the short and extended refinement strategies can bring customized attributes and easily deform the style of 3D assets, thus enabling flexible and controllable generation.

### C.3 COMPOSITIONAL GENERATION

We present more generated compositional 3D assets by our Layout-Your-3D in Fig. 17, here we only show results generated with our short instance-wise refinement setting (~12 mins in all). We can see that Layout-Your-3D can generate compositional assets, in which 3D instances can be in all 'granularity', from smaller ones like 'blue sunglasses', to bigger ones like 'A box', 'A toy pyramid'. This can be another strong evidence, showcasing that our method has great advantages in the aspect of controlling 3D generation.

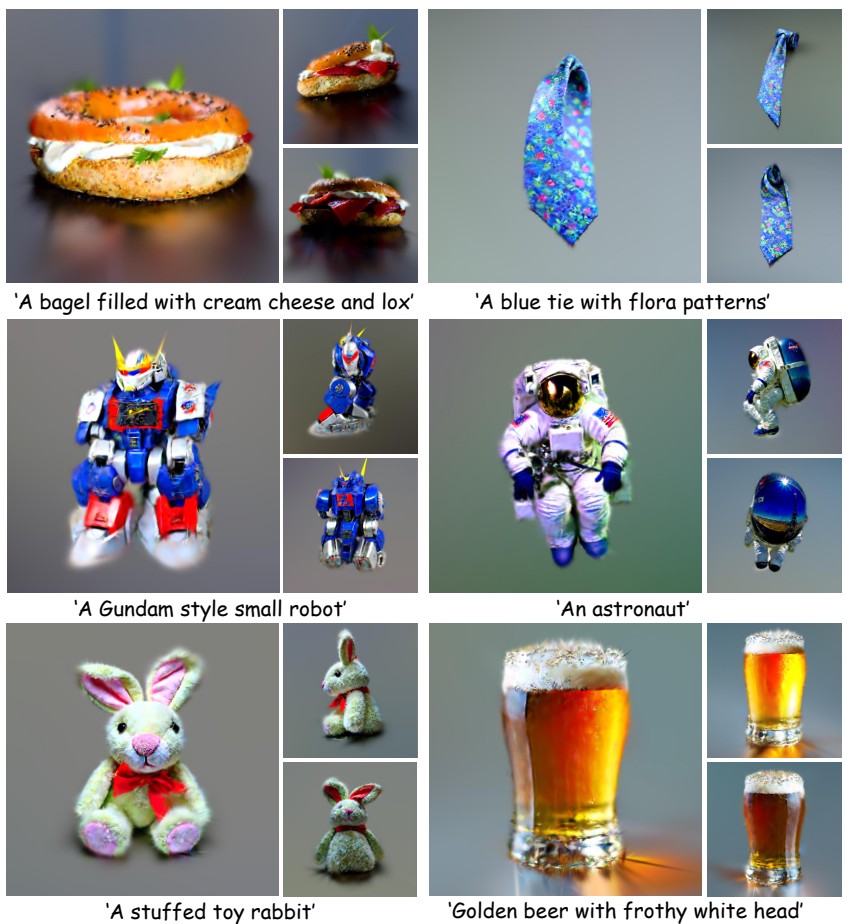

Figure 15: More results on the single object generation with our extended refinement strategy.

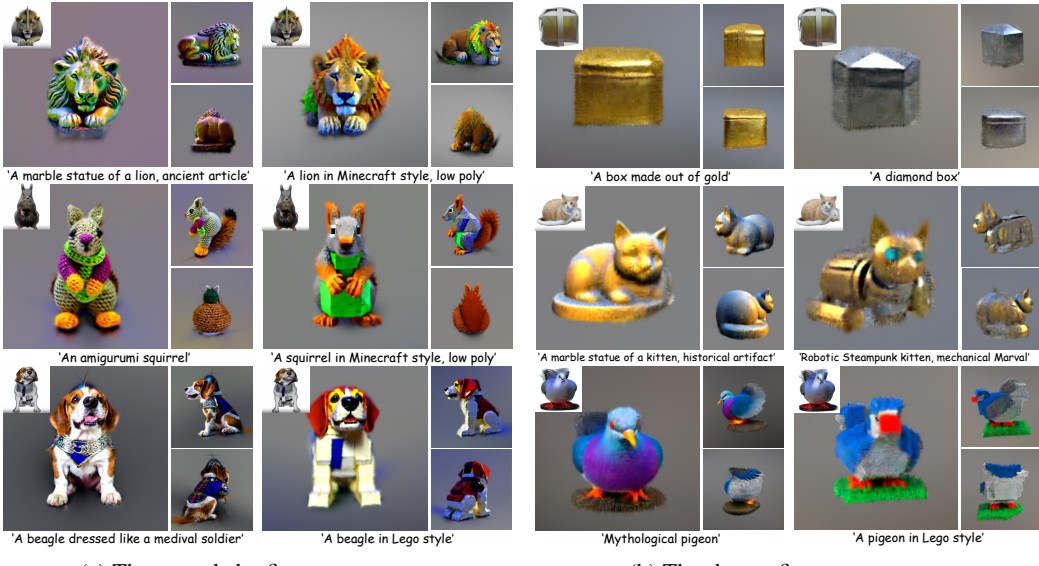

(a) The extended refinement strategy.

(b) The short refinement strategy.

Figure 16: Additional visualization results on instance generation with custom text prompts. We show results generated with both the short and extended instance-wise refinement strategies. White images on the top left corner of each figure are the original coarse 3D instances produced by LGM.

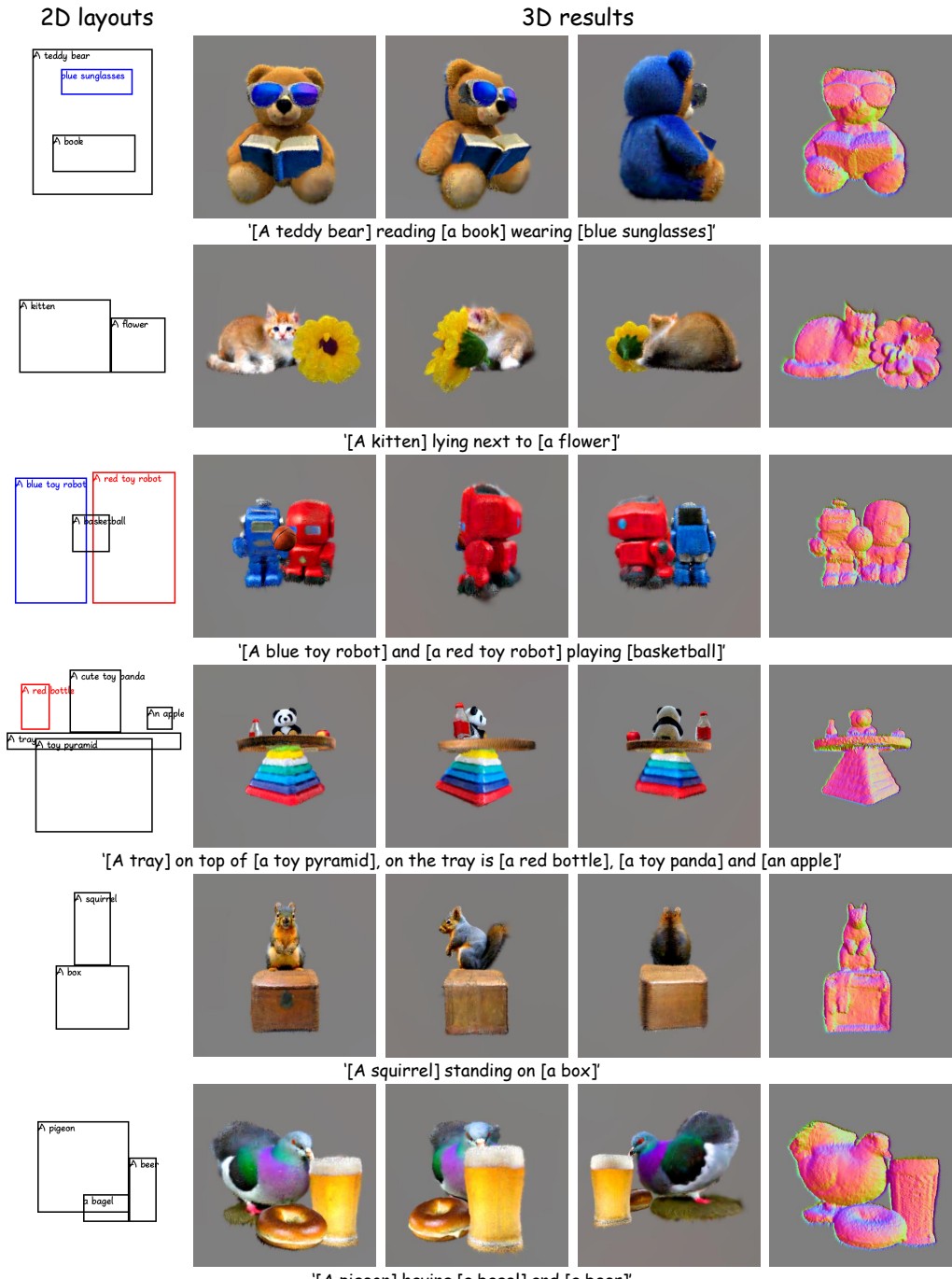

Figure 17: More visualization results generated with our $Comp20$ text prompts and layouts.

# D ADDITIONAL RESULTS & COMPARISONS

## D.1 LAYOUT INITIALIZATION

In this section, we provide more visualization results on the effectiveness of our 3D layout initialization. We compare different 3D layouts generated with LLM inference (Gao et al., 2024; Zhou et al., 2024b) and our strategy in Fig. 18. LLM-generated 3D layouts cannot align with the input text prompts well, thus posing significant optimization problems in the following refinement pro-

cess. In comparison, our 3D layout initialization strategy benefits from deployed 2D models, and can generate both precise and reasonable layouts. These comparisons reflect the motivation of using 2D blueprint for 3D layout initialization, instead of generating directly from LLMs.

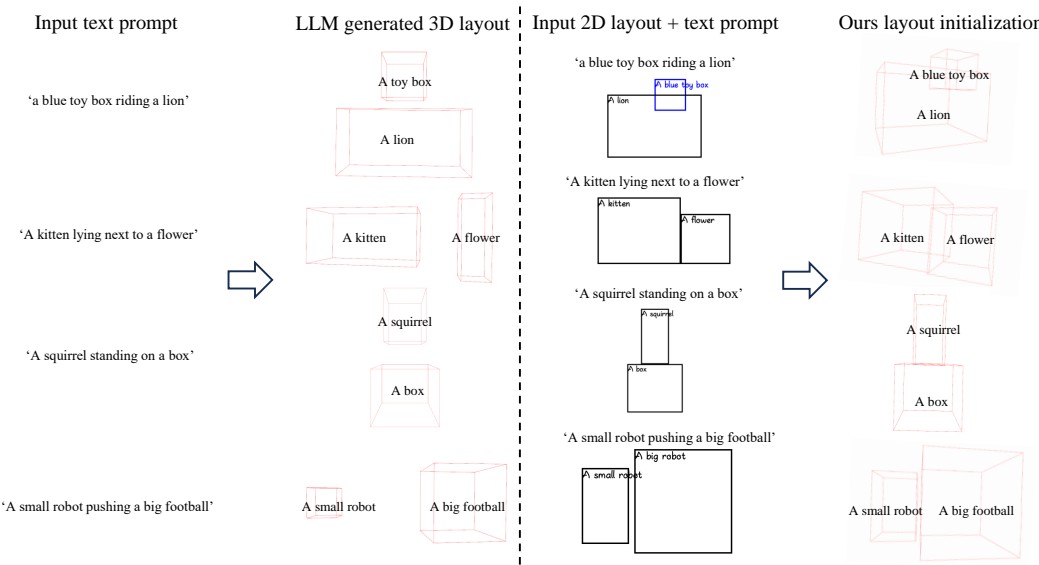

Figure 18: Visualization results of our initialized 3D layout, together with a comparison with LLM-generated coarse 3D layout boxes.

## D.2 COMPARISON WITH OTHER WORKS

Since ComboVerse has not released the official implementations, we compare Layout-Your-3Dwith the unofficial version in our experiments. For the unofficial implementation, we use the same initialization strategy as our Coarse 3D Generation Stage, instead of the one presented in ComboVerse. The comparisons are shown in Fig. 19. The left images for each prompt are 3D scenes generated by our implemented ComboVerse and the right ones are results generated with Layout-Your-3D.

Moreover, we compare the results generated with the same prompt by GALA3D and our Layout-Your-3D in Fig. 20. The figure shows that GALA3D tends to produce unreasonable 3D scenes when given poorly initialized 3D layout boxes. It is worth mentioning that GALA3D takes about 3 hours to generate one scene with 3 instances, which can be inefficient compared to our Layout-Your-3D (12 minutes).

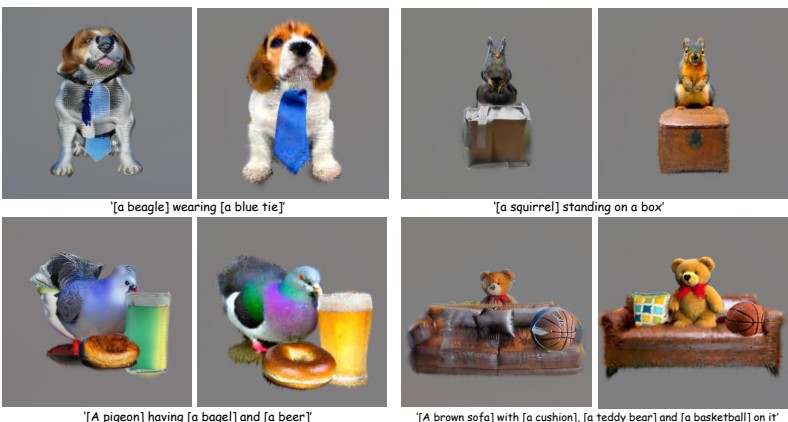

Figure 19: Comparison with ComboVerse.

*GALA3D*    *Layout-Your-3D*

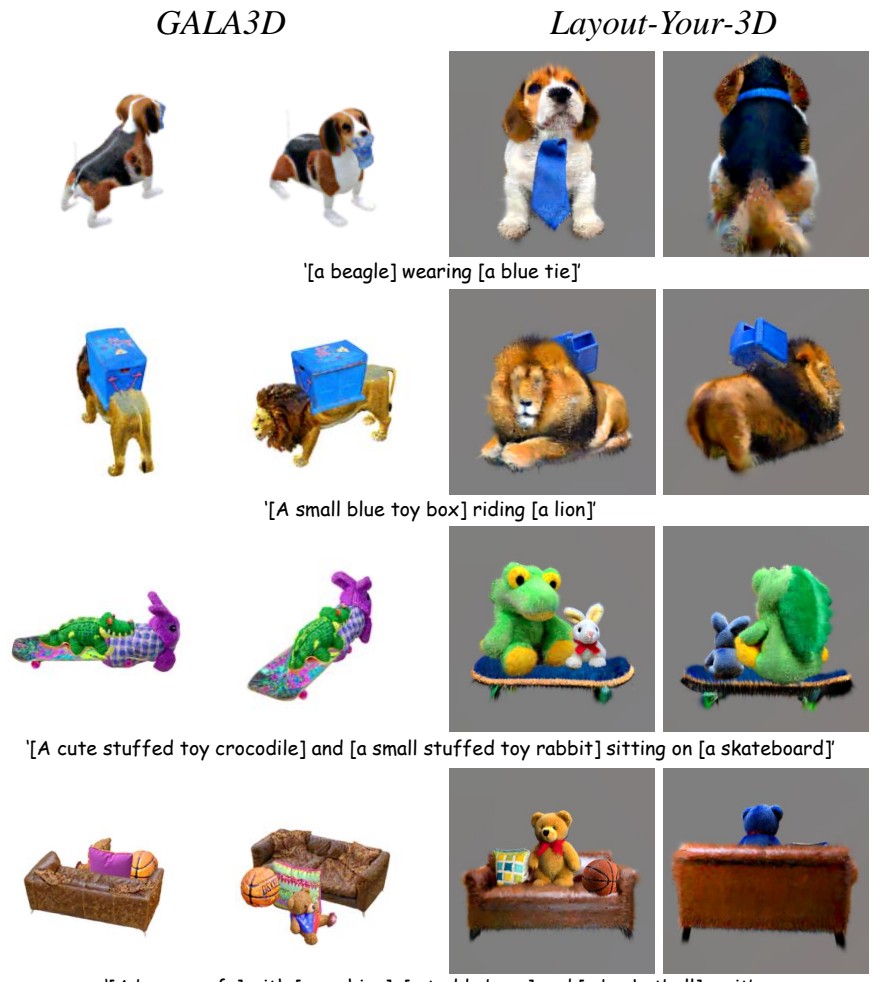

'[a beagle] wearing [a blue tie]'

'[A small blue toy box] riding [a lion]'

'[A cute stuffed toy crocodile] and [a small stuffed toy rabbit] sitting on [a skateboard]'

'[A brown sofa] with [a cushion], [a teddy bear] and [a basketball] on it'

Figure 20: Comparison with GALA3D.

### D.3 ABLATION ON ROTATION ESTIMATION

We conduct ablation studies on the effect of visual foundation models used when extracting features for coarse rotation estimation in Tab. 8. The 'Average error' term in the table represents the average deviation degrees for all instances in the validation set, we calculate and estimate the error in degrees manually. As shown in the table, the deployed visual foundation models significantly reduce rotation error, compared to scenarios where rotation estimation is not applied, thus validating the effectiveness of our coarse rotation estimation strategy.

Table 8: Ablation study on the foundation models used when extracting features for coarse rotation estimation.

| Method | No estimation | CLIP (Radford et al., 2021) | DINOv2 (Oquab et al., 2023) |
|---|---|---|---|
| Average error | 18.9° | 10.8° | 5.5° |

Table 9: Detailed prompt list of our proposed $Comp20$ validation set, including overall text prompt $Y_B$ and 2D layout set $B$. We provide both the user-given 2D layout set and the LLM-generated one.

| Index | Text Prompt |
|---|---|
| 1 | [A black bottle], with [a white bottle] on one side and [a blue bottle] on the other |
| User | [24, 136, 168, 424], [152, 72, 336, 424], [344, 104, 504, 424] |
| LLM | [200, 200, 280, 360], [100, 200, 180, 360], [300, 200, 380, 360] |
| 2 | [A teddy bear] reading [a book] wearing [blue sunglasses] |
| User | [88, 40, 168, 192], [144, 24, 120, 128], [168, 96, 112, 168] |
| LLM | [104, 64, 168, 200], [152, 0, 128, 168], [182, 112, 88, 184] |
| 3 | [A squirrel] standing on [A box] |
| User | [204, 51, 51, 0], [153, 0, 102, 179] |
| LLM | [190, 150, 222, 330], [172, 330, 340, 442] |
| 4 | [A small blue toy box] riding [a lion] |
| User | [0, 153, 102, 0], [102, 204, 153, 153] |
| LLM | [0, 140, 80, 10], [120, 200, 168, 168] |
| 5 | [A small robot] pushing [a big football] |
| User | [51, 153, 179, 102], [194, 102, 204, 128] |
| LLM | [75, 250, 225, 200], [225, 275, 200, 200] |
| 6 | [A pigeon] having [a bagel] and [a beer] |
| User | [102, 102, 358, 358], [230, 307, 358, 384], [358, 204, 435, 384] |
| LLM | [40, 88, 232, 416], [240, 360, 368, 416], [376, 168, 488, 416] |
| 7 | [An astronaut] sitting on [a sofa] |
| User | [153, 102, 384, 409], [51, 153, 460, 384] |
| LLM | [165, 130, 347, 392], [80, 100, 432, 372] |
| 8 | [A kitten] lying next to [a flower] |
| User | [51, 153, 307, 358], [307, 204, 460, 358] |
| LLM | [95, 218, 269, 492], [297, 224, 417, 382] |
| 9 | [A beagle] wearing [a blue tie] |
| User | [117, 51, 394, 460], [230, 204, 281, 435] |
| LLM | [88, 96, 424, 464], [208, 288, 312, 464] |
| 10 | [A teddy bear] driving [a toy car] |
| User | [40, 184, 472, 384], [215, 119, 335, 326] |
| LLM | [160, 150, 352, 406], [140, 320, 372, 470] |
| 11 | [A lego man] riding [a horse], the horse is wearing [a blue hat] |
| User | [ 96, 64, 288, 352], [39, 181, 481, 463], [367, 152, 463, 240] |
| LLM | [150, 200, 280, 400], [120, 250, 390, 450], [220, 150, 290, 220] |
| 12 | [A blue toy robot] and [a red toy robot] playing [basketball] |
| User | [39, 64, 240, 416], [256, 48, 488, 416], [199, 167, 304, 272] |
| LLM | [50, 200, 190, 440], [320, 200, 460, 440], [225, 280, 285, 340] |
| 13 | [A brown sofa] with [a cushion], [a teddy bear] and [a basketball] on it |
| User | [32, 256, 480, 432], [200, 120, 328, 344], [72, 224, 168, 360], [360, 256, 456, 352] |
| LLM | [[20, 180, 492, 412], [140, 230, 240, 310], [250, 220, 340, 330], [380, 250, 460, 330] |
| 14 | [A dog] taking [a boat] |
| User | [24, 256, 488, 384], [160, 104, 336, 304] |
| LLM | [186, 150, 326, 290], [50, 250, 462, 780] |
| 15 | [A furry fox's head] with [a cupcake] on it, wearing [red sunglasses] and have [a lego block] on the nose |
| User | [40, 64, 480, 512], [168, 72, 336, 216], [64, 256, 440, 376], [168, 264, 256, 407] |
| LLM | [176, 128, 336, 288], [224, 88, 288, 152], [200, 160, 312, 200], [240, 208, 272, 240] |
| 16 | [A cute stuffed toy crocodile] and [a small stuffed toy rabbit] on [a skateboard] |
| User | [40, 360, 472, 432], [72, 72, 296, 376], [296, 200, 424, 376] |
| LLM | [40, 304, 480, 432], [80, 112, 280, 344], [296, 176, 440, 336] |
| 17 | [A toy dragon] reaching for [a cola] on [a wooden cabinet] |
| User | [256, 192, 456, 416], [32, 104, 248, 416], [312, 80, 392, 200] |
| LLM | [101, 179, 289, 400], [316, 233, 394, 471], [40, 370, 472, 500] |
| 18 | [A tray] on top of [a toy pyramid], on the tray is [a red bottle], [a toy panda] and [an apple] |
| User | [96, 208, 424, 471], [56, 56, 136, 184], [408, 120, 480, 184], [16, 192, 504, 240], [192, 16, 336, 192] |
| LLM | [150, 300, 350, 500], [140, 260, 360, 320], [160, 100, 220, 260], [240, 130, 320, 260], [330, 120, 390, 180] |
| 19 | [A toy heart] and [a white feather] on [a balance scale] |
| User | [56, 136, 168, 248], [336, 192, 456, 240], [40, 224, 480, 384] |
| LLM | [140, 220, 240, 320], [272, 235, 352, 305], [100, 150, 412, 362] |
| 20 | [A green pumpkin] and [a big round orange] under [an umbrella] |
| User | [48, 200, 288, 424], [280, 272, 448, 416], [8, 120, 504, 224] |
| LLM | [112, 290, 240, 418], [272, 290, 400, 418], [110, 50, 402, 290] |

