# OpenReview forum: "Layout-your-3D: Controllable and Precise 3D Generation with 2D Blueprint"
_ICLR.cc/2025/Conference — ICLR 2025 Poster_

### Official Review · Reviewer_YpWR · 2024-10-16

**Soundness:** 3
**Presentation:** 2
**Contribution:** 2
**Rating:** 5
**Confidence:** 3

**Summary:**

This paper uses a 2D layout as guidance and proposes a Collision-aware Layout Refinement to optimize 3D layouts, effectively achieving compositional 3D generation with plausible object interactions.

**Strengths:**

- The paper is rich in content, and the proposed method successfully accomplishes the task set by its motivation.
- The writing is clear, making it easy to understand the authors' research methods.

**Weaknesses:**

- Over-reliance on existing methods: The engineering aspect of the method is too pronounced. Aside from the newly introduced loss functions, much of the technique is a stack of existing methods, including SAM, ControlNet-tile, LGM, GeoWizard, DINO, etc. The use of LLM to assist in generating the 2D layout is also an existing strategy. Such extensive reliance on pre-existing methods can overshadow the key contribution of the paper. Additionally, I would like to ask: since LLMs are already in use, why not have the LLM or MLLM directly predict the 3D layout and then apply your Collision-aware Layout Refinement for 3D layout refinement? This would simplify the initial stage, reduce reliance on existing methods, and highlight the contribution of your technique, making the overall framework more elegant and concise.

- Limited contribution: The task goal overlaps significantly with many existing compositional 3D scene generation methods (GALA3D, graphdreamer, dreamscene， progress3D...), making the problem being addressed less novel. Furthermore, as mentioned earlier, many critical components are stacked from existing methods. Although the Feature-level Reference Loss and Tolerant Collision Loss are novel, their impact seems less substantial compared to the other parts, as seen in the visual results in Figure 8. As a suggestion for improvement, it might be better to focus on complex single-image-to-3D reconstruction instead of compositional 2D layout + text-to-3D generation. You could use multimodal LLMs to decompose objects from a complex, unclear image for reconstruction and then refine and combine them into a 3D scene. This shift could substantially enhance the novelty and contribution of the paper.

- Insufficient comparison in experiments: In the supplementary materials, methods like GALA3D and DreamScene are mentioned. Why aren’t there comparisons with compositional 3D scene generation methods in the main text, such as GALA3D, GraphDreamer, DreamScene, Progressive3D, or more recent works like The Scene Language? Even if comparing with the very recent papers isn't necessary, it would be more reasonable to compare your work with three or so methods that are highly relevant to your contribution, such as GALA3D, GraphDreamer, and DreamScene. This could also help improve the previous point about novelty. Additionally, based on my own experience, DreamFusion should be faster than LucidDreamer—on my computational resources, the timing is approximately 30 minutes vs. 50 minutes, differing from the 1 hour vs. 1 hour stated in your paper. Although a minor issue, it raises some doubts about the accuracy of the reported experimental results.

**Questions:**

- My main concerns are discussed in the Weaknesses section. I can see from the paper that the authors have put a lot of effort into creating a valuable contribution, and despite my doubts about the contribution, experimental results, and design of the method, I still rate the paper as borderline reject (5). I believe the improvements needed cannot realistically be achieved during the rebuttal stage, and this may be due to time constraints in preparing the submission. I hope the authors can improve the method’s effectiveness and rewrite the contribution section to better highlight their approach, so that the paper can be successfully accepted at top conferences like **CVPR, ICML, or ICCV**.

- One final note: while I don't mean to discourage the authors, I have limited expectations for changes in the rebuttal, as the scope of improvement I am suggesting is quite large. Therefore, **it might not be necessary to spend too much time adding experiments during rebuttal for me**. I wish the authors all the best with their research.

---

> ### Author Response · Authors · 2024-11-20
> **Response to Reviewer YpWR**
>
> We thank you for your valuable suggestions and comments.
>
> We will address all the concerns point by point.
>
> **Weakness 1**: Over-reliance on existing methods.
>
> **Answer**: Please refer to **general comment 2**. Though we use some existing methods in our pipeline, the way how we combine these methods is novel and well-motivated. As for the usage of LLM/MLLMs, please refer to **general comment 3**, we provided some examples to illustrate the disadvantages of using LLMs to infer 3D layouts.
>
> ***
>
> **Weakness 2**: Limited contribution.
>
> **Answer**: Please refer to **general comment 2**. It is worth noting that there are **already** some works focusing on Compositional Image-to-3D reconstruction, which have demonstrated promising results. However, these kinds of approaches have limited application scenarios, which is why we chose Text-to-3D generation as our final goal. Moreover, using LLMs as guidance often produces unreasonable results that require tedious optimization or may even negatively impact the entire generation process. We also note that generating 3D assets automatically with text prompts from users is a significant part of our work, but not the only one. We would like to emphasize the flexibility of 2D layouts for controllable 3D generation, which can be a promising application in the future of 3D generation. 'Controllable' and 'Precise', are also considered as two targets in our work.
>
> ***
>
> **Weakness 3**: Insufficient comparison in experiments.
>
> **Answer**: Please refer to **general comment 1**. As suggested, we will add GraphDreamer to our paper in the revision (marked in red). We note that since Progressive3D mainly focuses on editing an existing 3D model, which is a different setting from ours, thus we do not compare with Progressive3D in our revision.
> For the experimental results, it is true that we made a typo on the time consumption, but in our test, it takes 45 minutes on one single RTX 6000 Ada Gen GPU card to generate one single asset for DreamFusion. Note that we apply the implementation on threestudio for our experimental results, which might be slightly different from the original one. We have fixed this typo in the revision.
>
> Hope our response can remove your doubts and help you have a better understanding of our work!

---

> ### Author Response · Authors · 2024-11-23
> **Follow up**
>
> Dear Reviewer YpWR:
>
> Thank the valuable and detailed comments on our paper, which provided insights that can help us revise our work.
>
> We have provided a response and a revised paper, hope they could address your concerns. We would like to know if there are more concerns about the contribution and motivation of our work. Also, we could provide more analyses on some aspects like the model selection for layout generation instead of MLLM/LLMs, which might help you better understand our paper. Your invaluable feedback and suggestions are greatly welcomed to help us refine our work.
>
> Thank you again for your devotion to the review. If all the concerns have been successfully addressed, please consider raising the scores after this discussion phase.
>
> Best,
>
> Submission#2873 Authors

---

> > ### Comment · Reviewer_YpWR · 2024-11-23
> > **Response to Authors**
> >
> > Thank you for your response. I can see that you put in great effort to address each question carefully. However, since GALA3D has already adopted the strategy of using LLM to plan layouts, the novelty of this paper appears to be less significant. Additionally, the paper relies heavily on existing methods. After careful consideration, I have decided not to raise my rating. That said, in recognition of your diligent rebuttal, I have decided to lower my confidence instead. I wish you all the best with your work. As an additional note, for high-quality 3D synthesis, I personally prioritize the quality of the generated results over speed. Therefore, I suggest considering further improvement of the quality in Fig. 5.

---

> > > ### Author Response · Authors · 2024-11-23
> > >
> > > Thank you for your valuable reply.
> > >
> > > As we have mentioned above, the utilization of LLMs to plan layouts is not a significant contribution of our work, it is only a simple way to obtain 2D layouts for fair comparison with other Text-to-3D generation works, and make the whole pipeline automatic. Lifting from 2D layouts to reasonable 3D layouts is a non-trivial task, and this could be part of our contribution. Hope this will help you change your assessment a little bit.
> > >
> > > Finally, thank you again for your detailed comments on how to refine our paper!

---

### Official Review · Reviewer_PaDe · 2024-10-18

**Soundness:** 3
**Presentation:** 2
**Contribution:** 2
**Rating:** 5
**Confidence:** 3

**Summary:**

the work trying to generate a 3D complex scene based on a textual prompt. they start by generating a 2D image of the scene using a 2D text-to-image generator, they extract 2D boxes for each instance.
Next, they create each object in a 3D instance and position them according to the boxes.
the final step is SDS which fine-tune the positions for natural placement, specifically in a collision-free manner.

[1] Yang, Yue, et al. "Holodeck: Language guided generation of 3d embodied ai environments." Proceedings of the IEEE/CVF Conference on Computer Vision and Pattern Recognition. 2024
[2] Rahamim, Ohad, et al. "Lay-A-Scene: Personalized 3D Object Arrangement Using Text-to-Image Priors." arXiv preprint arXiv:2406.00687 (2024).

**Strengths:**

generating complex scenes is a difficult task today.
this work tries to solve this task, into stages to deal with each part separately.

**Weaknesses:**

1. no novelty in the approach, you concatenated known strategies while trying to solve this problem.
2. can it be solved using previous methods like [1]
3. Besides the bear and the book, I can't see any interaction between objects. can you propose how to create interactions? (like the bird drinking the beer)

**Questions:**

1. why do the bird, bear, and Bagle look different in each example?
2. what happens if the generated 2D images don't contain all the specified objects? it does happen a lot with the generation of complex scenes. like what is shown in [2]

---

> ### Author Response · Authors · 2024-11-20
> **Response to Reviewer PaDe**
>
> Thank you for your suggestions and review.
>
> We will address all of your concerns and questions in the following response.
>
> **Weakness 1**: no novelty in the approach, you concatenated known strategies while trying to solve this problem.
>
> **Answer**: We respectfully disagree with the reviewer on this point. Please refer to the **general comment 2**, we have emphasized our novelty and motivation again in the comment.
>
> ***
>
> **Weakness 2**: can it be solved using previous methods like Holodeck.
>
> **Answer**: We emphasize that Holodeck operates under a fundamentally different setting from ours. Specifically, Holodeck aims at retrieving 3D objects from an existing 3D dataset and fitting them in locations generated by LLMs. In contrast, our work focuses on generating target 3D assets with text prompt and 2D layout input, in which the 3D objects are not predefined and can be customized as shown in **Fig. 10** and **11**. Furthermore, in **general response 3**, we show that LLMs tend to produce inferior 3D layouts.
>
> ***
>
> **Weakness 3**: Besides the bear and the book, I can't see any interaction between objects. can you propose how to create interactions?
>
> **Answer**: Please refer to **Fig. 5** for further clarification regarding your concern. In our paper, the word 'interaction' has many aspects, including relative locations, scales, poses, and even motions of different instances. In **Fig. 5**, for each prompt, our method can compose the instances with proper 'interactions', like a beagle 'wearing' a blue tie, some stuffed toys 'sitting on' a skateboard and some stuff placed right 'on' the sofa.
>
> By 'interaction', if the reviewer indicates specific motions, we would like to point to the stuffed toys 'sitting' on a skateboard in **Fig. 5**, the astronaut _'sitting'_ in **Fig. 15**, and kitten _'lying next to'_ a flower in **Fig. 17**. These words are commonly recognized as motions in our daily lives.
>
> As for how to create interactions, we recommend you to look at **L322-L323** (marked in orange). We first create the desired interactions between objects in the generated reference image and then retain these interactions in the following reconstruction and refinement process.
> Please let us know if you have any further concerns on this point.
>
> ***
>
> **Question 1**: why do the bird, bear, and Bagle look different in each example?
>
> **Answer**: Please check out two types of 2D layout inputs we provided in **Fig. 7** and **L364-L366**. Since 2D diffusion models tend to produce varying results based on **different 2D input layouts** (*user-provided* or *LLM-generated*) and random seeds, the resulting 3D instances may appear different in the paper since the 3D results are first reconstructed from the 2D images. In the instance-wise refinement section, we propose **two strategies** for short and extended refinement. Each strategy utilizes **a different guidance model** (i.e., DeepFloyd and Stable Diffusion), and switching between guidance models during this process can also lead to variations in the results.
>
> ***
>
> **Question 2**: what happens if the generated 2D images don't contain all the specified objects? it does happen a lot with the generation of complex scenes. like what is shown in Lay-A-Scene.
>
> **Answer**: We ran 1000 cases to test the robustness of layout-conditioned image generation, and 99.8% of these examples contain all of the specified objects. Based on the capabilities of current layout-guided 2D generation models and the experiments we add, it **does not** qualify as a failure case in our discussion. This is also our motivation to resort to 2D layouts instead of 3D layouts. Furthermore, if the input image depicts two objects where one is fully occluded by the other, none of the existing 3D generation methods could generate the fully occluded object.
>
> Hope our response will alleviate your concerns.

---

> > ### Comment · Reviewer_PaDe · 2024-11-24
> >
> > Thank you very much for the answers.
> > After observing more interaction, I decided to raise my rating.

---

> > > ### Author Response · Authors · 2024-11-24
> > >
> > > Thank you very much for your careful consideration. It is good to see that our comments could address your concerns. If you have more questions or concerns, please do not hesitate to raise them. Your valuable suggestions can be helpful for us to refine our paper.

---

> ### Author Response · Authors · 2024-11-23
> **Follow up**
>
> Dear Reviewer PaDe:
>
> Thank the valuable comments on our paper, which provided insights that can help us revise our work.
>
> We have provided a response and a revised paper, hope they could address your concerns. Also, we would like to know if there are more concerns about the content of the paper, like some settings we adopted during implementation or differences between other works. Your invaluable feedback and suggestions are greatly welcomed to help us better refine our work.
>
> Thank you again for your devotion to the review. If all the concerns have been successfully addressed, please consider raising the scores after this discussion phase.
>
> Best,
>
> Submission#2873 Authors

---

### Official Review · Reviewer_miqw · 2024-11-02

**Soundness:** 3
**Presentation:** 3
**Contribution:** 2
**Rating:** 6
**Confidence:** 3

**Summary:**

This paper tries to address the problem of lay-out controlled 3D generation task.
Interestingly, it relies on 2D layout instead of 3D layouts, and the 2D layout could be generated by LLM given a text prompt.
With this 2D Layout, it first generate a reference image with an off-the-shelf 2D-layout image generation model, and then try to reconstruct the 3D model and layout from the reference image.
This generated 3D model is coarse and may not very accurate, and this paper refines the generated model with SDS and feature loss, as well as collision-tolerant loss.
This paper proposes a very interesting pipeline using many off-the-shelf components, and it achieves good results

**Strengths:**

1. This paper proposes an interesting pipeline by combining several (off-the-shelf) components in a novel way. Getting 2D reference images from layout and trying to reconstructing 3D models from them leads a interesting 3D reconstruction task, and this paper provides a reasonable solution for it, by utilizing the inpainting, DINO feature distance and off-the-shelf image generation model.
2. This paper involves several intesting losses like collision-tolerant loss, feature-level loss, and SDS with adjusted timesteps.
3. Results are good. and the paper is easy to read and follow.

**Weaknesses:**

1. For baselines, it would be more convincing to seriously compare against models with 3D layout control like ComboVerse or [Compositional 3D Scene Generation using Locally Conditioned Diffusion]. The current paper only shows a single scene comparison against ComboVerse in Figure 7, which is insufficient to demonstrate the advantages of using 2D layout and the proposed method. More rigorous evaluation, such as those shown in Tables 1 and 2, along with a user study, would be needed for such comparisons.

2. After reconstructing individual 3D models from 2D images, it is critical to align the pose (6D pose) of the reconstructed 3D model with the image appearance. The proposed method achieves this by randomly sampling many views of the 3D model and comparing them against the reference image by computing the DINO feature distance. However, I am curious about the accuracy of this method, as the rendered image could differ significantly from the reference image even with perfect poses if the distance (object scale) is not accurate. It would be more convincing to show results demonstrating the accuracy of this method.

3. Regarding the reconstruction of objects from reference images, how is the mask/segmentation obtained for each object? Is this process automated, or does it require manual intervention? This question also appears in the following inpainting and pose alignment sections. Do we need to manually specify which object to inpaint or segment during this process or not?

**Questions:**

1. For clarification, when inputing the text prompts, what's the prompt for LLM to generate 2D layouts? Also, do you mark the object in bracket like text prompts in figure 5 or LLM is supposed to decide which objects should be created?

2. The proposed method uses many off-the-shelf algorithm and components like LLM generating 2D layouts, reference image generation, pose alignment. I am curious about the successful ratio of the whole process, and some potential failure cases.

---

> ### Author Response · Authors · 2024-11-20
> **Response to Reviewer miqw**
>
> Thank you for your valuable comments and suggestions.
>
> Below, we will address each of your questions and doubts one by one.
>
> **Weakness 1**: For baselines, it would be more convincing to seriously compare against models with 3D layout control like ComboVerse or [Compositional 3D Scene Generation using Locally Conditioned Diffusion]. The current paper only shows a single scene comparison against ComboVerse in Figure 7, which is insufficient to demonstrate the advantages of using 2D layout and the proposed method. More rigorous evaluation, such as those shown in Tables 1 and 2, along with a user study, would be needed for such comparisons.
>
> **Answer**: It is worth noting that we mainly focus on Text-to-3D generation instead of Image-to-3D reconstruction in this work. Moreover, since both ComboVerse and Locally Conditioned Diffusion did not release their official implementations, we are unable to compare with them. However, since we have implemented some parts of ComboVerse, we present the comparison between our work and the unofficial version of ComboVerse in **Appendix D.2** and **Fig. 19**.
>
> ***
>
> **Weakness 2**: After reconstructing individual 3D models from 2D images, it is critical to align the pose (6D pose) of the reconstructed 3D model with the image appearance. The proposed method achieves this by randomly sampling many views of the 3D model and comparing them against the reference image by computing the DINO feature distance. However, I am curious about the accuracy of this method, as the rendered image could differ significantly from the reference image even with perfect poses if the distance (object scale) is not accurate. It would be more convincing to show results demonstrating the accuracy of this method.
>
> **Answer**: As shown in Fig. 7 in the main paper, our coarse rotation estimation successfully aligns the 3D insances' poses with the reference image.
>
> Moreover, we conducted experiments on the accuracy of our coarse rotation estimation in **Tab. 8** and **Appendix D.3**. Specifically, we manually calculated the average error degrees of all instances in the *Compo20* validation set and provided two models for comparison. We hope these results could provide more insights to the successful rate of rotation estimation.
> As for the effects of distance, since we adjust the border ratio of instances in the rendered images during our experiments, we can assume that they share the same camera extrinsic. This helps mitigate the negative effects of distance (object scale) on our coarse rotation estimation.
>
> ***
>
> **Weakness 3**: Regarding the reconstruction of objects from reference images, how is the mask/segmentation obtained for each object? Is this process automated, or does it require manual intervention? This question also appears in the following inpainting and pose alignment sections. Do we need to manually specify which object to inpaint or segment during this process or not?
>
> **Answer**: When reconstructing instances, we feed the 2D layouts into SAM to obtain the instance masks. Please refer to 'Individual 3D Object Reconstruction' part (from **Line 203**, marked in orange) for more information. To avoid confusion, here we also take the centers of other 2D layout boxes as negative prompts when segmenting one instance. During the generation process, we **do not** incorporate any manual intervention, which is the same for the following inpainting and pose alignment process.

---

> ### Author Response · Authors · 2024-11-20
> **Response to Reviewer miqw**
>
> **Question 1.1**: For clarification, when inputing the text prompts, what's the prompt for LLM to generate 2D layouts?
>
> **Answer**: We use LLM-Grounded Diffusion [1] for 2D layout generation. The LLM prompt is as follows:
>
>     You are an intelligent bounding box generator. I will provide you with a caption for a photo, image, or painting. Your task is to generate the bounding boxes for the objects mentioned in the caption, along with a background prompt describing the scene. The images are of size 512x512. The top-left corner has coordinate [0, 0]. The bottom-right corner has coordinnate [512, 512]. The bounding boxes should not overlap or go beyond the image boundaries. Each bounding box should be in the format of (object name, [top-left x coordinate, top-left y coordinate, box width, box height]) and include exactly one object (i.e., start the object name with "a" or "an" if possible). Do not put objects that are already provided in the bounding boxes into the background prompt. Do not include non-existing or excluded objects in the background prompt. If needed, you can make reasonable guesses. Please refer to the example below for the desired format.
>
>     Caption: A realistic image of landscape scene depicting a green car parking on the left of a blue truck, with a red air balloon and a bird in the sky
>     Objects: [('a green car', [21, 281, 211, 159]), ('a blue truck', [269, 283, 209, 160]), ('a red air balloon', [66, 8, 145, 135]), ('a bird', [296, 42, 143, 100])]
>     Background prompt: A realistic landscape scene
>     Negative prompt:
>
>     Caption: A realistic top-down view of a wooden table with two apples on it
>     Objects: [('a wooden table', [20, 148, 472, 216]), ('an apple', [150, 226, 100, 100]), ('an apple', [280, 226, 100, 100])]
>     Background prompt: A realistic top-down view
>     Negative prompt:
>
>     Caption: A realistic scene of three skiers standing in a line on the snow near a palm tree
>     Objects: [('a skier', [5, 152, 139, 168]), ('a skier', [278, 192, 121, 158]), ('a skier', [148, 173, 124, 155]), ('a palm tree', [404, 105, 103, 251])]
>     Background prompt: A realistic outdoor scene with snow
>     Negative prompt:
>
>     Caption: An oil painting of a pink dolphin jumping on the left of a steam boat on the sea
>     Objects: [('a steam boat', [232, 225, 257, 149]), ('a jumping pink dolphin', [21, 249, 189, 123])]
>     Background prompt: An oil painting of the sea
>     Negative prompt:
>
>     Caption: A cute cat and an angry dog without birds
>     Objects: [('a cute cat', [51, 67, 271, 324]), ('an angry dog', [302, 119, 211, 228])]
>     Background prompt: A realistic scene
>     Negative prompt: birds
>
>     Caption: Two pandas in a forest without flowers
>     Objects: [('a panda', [30, 171, 212, 226]), ('a panda', [264, 173, 222, 221])]
>     Background prompt: A forest
>     Negative prompt: flowers
>
>     Caption: 'input text prompt'
>     Objects:
>
> [1]: LLM-grounded Diffusion: Enhancing Prompt Understanding of Text-to-Image Diffusion Models with Large Language Models
>
> ***
>
> **Question 1.2**: Also, do you mark the object in bracket like text prompts in figure 5 or LLM is supposed to decide which objects should be created?
>
> **Answer**: As for how to create 2D layouts when prompting LLMs, we simply let LLMs decide whether each object should be created and do not mark each object in bracket. This method succeeds in most cases in our trial, LLMs are sensitive to nouns and tend to make the right decisions on whether each object should be created or not (e.g. a beagle wearing a blue tie).
>
> ***
>
> **Question 2**: The proposed method uses many off-the-shelf algorithm and components like LLM generating 2D layouts, reference image generation, pose alignment. I am curious about the successful ratio of the whole process, and some potential failure cases.
>
> **Answer**: The successful ratio is relatively high in our experiments. For example, when generating for prompts in 'Compo20' validation set, we simply generate with one-shot for quantitative and qualitative results. Although the off-the-shelf methods will produce unreasonable results occasionally, the following optimization process will largely resolve these issues.
>
> For the potential failure cases, please refer to **Appendix A.2**, where one failure case is shown in **Fig. 12**.
> Hope our responses can address your concerns.

---

> ### Author Response · Authors · 2024-11-23
> **Follow up**
>
> Dear Reviewer miqw:
>
> Thank the valuable and detailed comments on our paper, which provided insights that can help us revise our work.
>
> We have provided a response and a revised paper, hope they could address your concerns. Also, we would like to know if there are more concerns (e.g., on the pose alignment process or more comparisons with 3D layout works). Your invaluable feedback and suggestions are greatly welcomed to help us better refine our work.
>
> Thank you again for your devotion to the review. If all the concerns have been successfully addressed, please consider raising the scores after this discussion phase.
>
> Best,
>
> Submission#2873 Authors

---

> > ### Comment · Reviewer_miqw · 2024-11-23
> > **Reply to rebuttal**
> >
> > Hi. Thanks for the rebuttal. It provided lots of valuable information.
> >
> > I agree with other reviews that the text/layout to 3D is not very novel and the overall novelty in this paper is mild.
> > But personally I think this paper has some interesting components like using LLM, 2D layout to 3D and others, and tbh, I am more interested in these components.
> >
> > Could you elaborate more about how could you solve the distance ambiguous  problem during pose estimation? I don't really get how the image broaders are related to the extrinsic.

---

> > > ### Author Response · Authors · 2024-11-25
> > > **Follow up on the comment**
> > >
> > > Dear Reviewer miqw:
> > >
> > > Thank you again for the devotion and detailed comments in the review process, they are helpful in the refinement process of our paper. Since the deadline for the rebuttal phase is approaching, we feel like it is necessary to follow up on your valuable comments.
> > >
> > > We have explained the rotation estimation strategy in detail, hope that will help you better understand the techniques used in our paper. If there are more points that you are interested in, please do not hesitate to raise them. We will try our best to fix them.
> > >
> > > If all the concerns have been successfully addressed, please consider raising the scores after this discussion phase.
> > >
> > > Best,
> > >
> > > Submission#2873 Authors

---

> > > ### Author Response · Authors · 2024-11-25
> > > **Looking forward to your reply**
> > >
> > > Dear reviewer miqw:
> > >
> > > Since the deadline for rebuttal is approaching, would you please consider our comments and check out if they could address your concerns? Your valuable suggestions are very important for us to refine our paper. And if our comments successfully address the raised points, please consider raising the score for our work.
> > >
> > > Thank you again for your devotion during this process.
> > >
> > > Submission#2873 Authors

---

> > > ### Author Response · Authors · 2024-11-28
> > > **Potential discussion and comments.**
> > >
> > > Dear Reviewer miqw:
> > >
> > > Since the rebuttal period has been extended, there is still some time for the discussion. Would you please check out our comments and see if your questions have been successfully addressed? If so, please consider increasing the score. Or if there are more questions and concerns, please directly raise them, and we will try our best to solve them.
> > >
> > > Best,
> > > Submission#2873 Authors

---

> > > ### Author Response · Authors · 2024-11-30
> > >
> > > Dear Reviewer miqw,
> > >
> > > Since the rebuttal period has been extended, there is still some time for the discussion. Please check out our comments and see if your questions have been successfully addressed. Also, If all the concerns have been addressed and our comments could make you more interested in our work, please consider increasing the score for our work. Thank you very much!
> > >
> > > Best,
> > >
> > > Submission#2873 Authors

---

> > > ### Author Response · Authors · 2024-12-02
> > >
> > > Dear Reviewer miqw,
> > >
> > > The due of the rebuttal phase is approaching. We have elaborated more on the rotation estimation part, please check out our comments and see if your questions have been successfully addressed. Also, If all the concerns have been addressed and our comments could make you more interested in our work, please consider increasing the score for our work. Thank you very much!
> > >
> > > Best,
> > >
> > > Submission#2873 Authors

---

> > > ### Author Response · Authors · 2024-12-02
> > >
> > > Dear Reviewer miqw,
> > >
> > > Sorry for repetitively posting comments. There is only less than one day before the rebuttal phase is due, would you please check out our comment and consider increasing the rating? We hope these comments could help you remove your concerns. Thank you very much for your time and devotion in the whole rebuttal phase!
> > >
> > > Best,
> > >
> > > Submission#2873 Authors

---

> > > ### Author Response · Authors · 2024-12-03
> > >
> > > Dear Reviewer miqw,
> > >
> > > With only several hours left, please check out our comment and consider increasing the rating.
> > >
> > > Thank you for your time and devotion during the whole rebuttal phase!
> > >
> > > Best,
> > >
> > > Submission#2873 Authors

---

> ### Author Response · Authors · 2024-11-24
>
> Thank you very much for the invaluable suggestions, and we are happy to know that you are interested in our work.
>
> We are the first few works to explore 2D layout to 3D, a completely different setting from other compositional 3D generation methods like GraphDreamer, GALA3D, and DreamScene (using LLMs to infer inaccurate and unreasonable 3D layouts). Combined with some proposed methods to solve this problem and the broad application scenarios, we think our contribution and novelty are clear.
>
> Here we explain the rotation estimation part in detail. Since the reconstruction model (LGM [1]) is trained under fixed rendering settings (fov, radius, etc.), it is necessary to adjust the input images to meet that setting (marked as **default setting** below) for better performance. Specifically, we could first segment out the instance in the input image, and then adjust the border ratio (the border between the instance's bounding rectangle and the edge of the image). When rendering, we still render under the same **default setting**. Thus when rendering at the same fov and radius, we can assume that the rendered images share the same camera settings.
>
> We adopt the same camera view setting as stated in LGM. We fix the camera radius to 1.5 and the field-of-view along Y axis to 49.1 degrees to form a spatial path on the sphere surface.
>
> Hope this comment could solve your concerns. Please let us know if you have more questions.
>
> [1] LGM: Large Multi-View Gaussian Model for High-Resolution 3D Content Creation.

---

### Official Review · Reviewer_tCHQ · 2024-11-03

**Soundness:** 3
**Presentation:** 2
**Contribution:** 3
**Rating:** 6
**Confidence:** 2

**Summary:**

- They enforce structural control of a 3D scene using a 2D layout. Their optimization process avoids collisions.
- Their method works by first generating a 2D image based on text description, then use SAM to segment each isntnace in the reference iamge, with in-painting done on occluded parts.
- They then use a large mulit-view gausian model to generate coarse 3D instances of objects within the scene.
- Each 3D instance is then arranged into 3D space by lifting the bounding boxes into 3D, done by optimizing the rotation according to similarity in the DINO space of the rendered images of the rotated instances.
- final two stages refine the layout according to collisions and refines the instance representations.

**Strengths:**

The papers shows a potential way to use many existing models together to generate compositional 3D scenes. The results show that the method produces reasonable visual reconstructions and geometries. The method seems well motivated, and the paper is decently written.

**Weaknesses:**

My concern is that this work lacks fundamental novelty, in that prior works like Compositional 3D Scene Generation using Locally Conditioned Diffusion (https://arxiv.org/abs/2303.12218), Towards Text-guided 3D Scene Composition (https://arxiv.org/pdf/2312.08885) and CG3D: Compositional Generation for Text-to-3D via Gaussian Splatting (https://arxiv.org/pdf/2311.17907) all present various ways to create 3D scenes using implicit 3D represntations in a compositional way.

There does not seem to be any comparisons to these works, or mention of them in the related works (though CG3D was mentioned in the collisions section, but not compared with in the experiments section). A comparison to contextualize this work seems very important, but lacking.

**Questions:**

Within the collision loss, is there any way to take into account the shape (covariance) of each gaussian in 3D space? My understanding is that it's only using the positions of the the Gaussians right now. Have you observed this to be a limitation?

---

> ### Author Response · Authors · 2024-11-20
> **Response to Reviewer tCHQ**
>
> Thank you for your valuable comments.
>
> We will address all the concerns point by point.
>
> **Weakness**: Lacking in fundamental novelty, no comparison with Layout Conditioned Diffusion, SceneWiz3D, CG3D.
>
> **Answer**: Please refer to the **general comment 1** for the reply of the weakness part 2. Since all of these works do not have official implementations, we are not able to provide comparisons and experiments on them. However, we have cited these works in the revision. For novelty and contribution, please refer to the **general comment 2**. Hope these comments could remove your doubts.
>
> ***
>
> **Question**: Within the collision loss, is there any way to take into account the shape (covariance) of each gaussian in 3D space? My understanding is that it's only using the positions of the the Gaussians right now. Have you observed this to be a limitation?
>
> **Answer**: The design of Collision loss mainly aims at applying penalty to those instances that have significant overlaps. Note that we count collision per instance, not per Gaussians. Each gaussian's shape can contribute very little to the whole object's shape and location. As a result, we have not observed this to be a limitation in the optimization process. Also, we provided an example in **Fig. 8** in the main paper.

---

> ### Author Response · Authors · 2024-11-23
> **Follow up**
>
> Dear Reviewer tCHQ:
>
> Thank the valuable comments on our paper, which provided insights that can help us revise our work.
>
> We have provided a response and a revised paper, hope they could address your concerns. Also, we would like to know if there are more concerns. Your invaluable feedback and suggestions are greatly welcomed to help us better refine our work.
>
> Thank you again for your devotion to the review. If all the concerns have been successfully addressed, please consider raising the scores after this discussion phase.
>
> Best,
>
> Submission#2873 Authors

---

> > ### Author Response · Authors · 2024-11-25
> > **Follow up comment for reviewer tCHQ**
> >
> > Dear Reviewer tCHQ:
> >
> > Thank you again for your valuable review, and sorry for repetitively sending comments.
> >
> > Since the deadline for the rebuttal phase is approaching, it could be important for us to follow up on your review and comments. We have provided a response and a revised paper, hope they could address your concerns. Your invaluable comments and feedback can be helpful when we refine our paper.
> >
> > Thank you again for your devotion to the review. If all the concerns have been successfully addressed, please consider raising the scores after this discussion phase. If there are still some concerns, please raise them and we will try to address as possible.
> >
> > Best,
> >
> > Submission#2873 Authors

---

> ### Author Response · Authors · 2024-11-25
> **Looking forward to your reply**
>
> Dear reviewer tCHQ:
>
> Since the deadline for rebuttal is approaching, would you please consider our comments and check out if they could address your concerns? Your valuable suggestions are very important for us to refine our paper. And if our comments successfully address the raised points, please consider raising the score for our work.
>
> Thank you again for your devotion during this process.
>
> Submission#2873 Authors

---

> ### Author Response · Authors · 2024-11-28
> **Potential discussion and comments.**
>
> Dear Reviewer tCHQ:
>
> Since the rebuttal period has been extended, there is still some time for the discussion. Would you please check out our comments and see if your questions have been successfully addressed? If so, please consider increasing the score. Or if there are more questions and concerns, please directly raise them, we will try our best to solve them.
>
> Best,
> Submission#2873 Authors

---

> ### Comment · Reviewer_tCHQ · 2024-11-28
>
> Thanks for providing the rebuttal, and apologies for getting to this later than I would have liked.
>
> The comparisons you show  in Figure 13, Figure 19, and Figure 20 address my concerns about comparisons to prior works, though Figure 19 seems like you'd have to label which one is ComboVerse and which one is your method (I assumed which is which, though).
>
> I think Figure 18 is the most convincing, because it shows that LLMs generating 3D layouts from input text suffers from many problems,  but that  a 2D layout *in addition* to a language prompt can ground the bounding box initialization enough to mitigate a lot of these problems.
>
> I've increased my rating for the paper.

---

> > ### Author Response · Authors · 2024-11-28
> > **Thank you for the comments**
> >
> > Dear reviewer,
> >
> > Thanks for the comments. In the final version of this paper, we will improve the label of Figure 19.
> >
> > Could you check whether you have increased the score on Openreview (as it seems to be the same as before)?
> >
> > Thank you,

---

> > ### Author Response · Authors · 2024-11-30
> >
> > Dear Reviewer tCHQ,
> >
> > Would you please check out the rating of our paper again? It is still the same as the original score.
> > Thank you very much for your valuable suggestions and review.
> >
> > Best,
> >
> > Submission#2873 Authors

---

> > ### Author Response · Authors · 2024-12-02
> >
> > Dear Reviewer tCHQ,
> >
> > Please check out the rating of our paper again, It is still the same as the original score.
> >
> > Thank you again for your valuable suggestions and review.
> >
> > Best,
> >
> > Submission#2873 Authors

---

> > > ### Comment · Reviewer_tCHQ · 2024-12-02
> > >
> > > It's been edited -- thanks for the reminder.

---

> > > > ### Author Response · Authors · 2024-12-02
> > > >
> > > > Thank you very much! Good to see that your concerns have been addressed.

---

### Author Response · Authors · 2024-11-20
**General comments to all the reviewers**

Dear reviewers,

We thank you for the valuable reviews and suggestions. We first address some common questions raised by reviewers.

## 1. Experiments and comparison with other compositional Text-to-3D generation methods (Reviewer tCHQ, miqw, YpWR)
Reviewers suggest to include comparisons with other Compositional 3D generation works, including Locally Conditioned Diffusion, SceneWiz3D, CG3D, ComboVerse, GALA3D, DreamScene, and GraphDreamer. Note that among all of these works, only GALA3D, DreamScene and GraphDreamer have released their official implementations. As a result, we may not be able to compare to other methods mentioned by reviewers. We added all references suggested by reviewers in **Line144-146** in the revised manuscript. Next, we compare the proposed method with GALA3D, DreamScene and GraphDreamer.

For **GALA3D**, the current publicly available code does not include the finetuned ControlNet. As a result, we cannot carry out the layout fine-tuning stage. To make the comparison as fair as possible, we reached out to the authors of GALA3D, they explained that the ControlNet is specially finetuned for indoor room layout, thus it will not affect the final results of common input prompts used in our paper. Therefore, we use the publicly available code of GALA3D and show the comparison in **Appendix D.2** and **Fig. 20**.

**DreamScene** aims at generating individual 3D objects and fitting them into fixed 3D layouts. However, we **do not** compare with DreamScene since it does not include layout refinement and cannot produce reasonable results that contain certain interactions between objects.

As suggested, we have included comparisons with **GraphDreamer** in the revised manuscript (**Fig. 5** and **Tab. 1**).

&nbsp;
## 2. Novelty and Contributions (Reviewer tCHQ, PaDe, YpWR)
We would like to clarify the novelty and contribution of this work.

- First, we exploit easily-accessible and accurate 2D layouts as blueprint for controllable and precise 3D generation. As stated in **L081-L082** in the manuscript (marked in orange color), we leverage 2D layout to provide an easily-accessible and accurate guidance for controllable and precise 3D generation. This setting differs from existing compositional text-to-3D generation methods, which rely on Large Language Models (LLMs) to generate imprecise 3D layouts that lead to subpar outcomes and time-consuming optimization processes (**Fig. 18** in **Appendix D.1**). This is our main motivation to leverage the strength and performance of 2D layout methods and lift the results to 3D.

- Second, we emphasize that generating 3D assets from 2D layout is a challenging task. To address this, we meticulously design a method that incorporates several novel modules, including: **a)** a 3D layout initialization module that predicts plausible object geometry, appearance, location as well as rotation, **b)** a refinement stage that improves not only individual object appearance but also their interactions, using a novel collision-aware constraint. Consequently, our method not only generates better results, but also significantly reduces processing time (minutes vs. hours) compared to existing approaches.

- Third, our work extends beyond text-to-3D generation. As illustrated in our paper (**Fig. 10, 11**), our pipeline also supports additional applications such as instance-wise customization and editing, offering greater control and flexibility in the generation process.

&nbsp;
## 3. Explain why not use LLM/MLLM to infer 3D layouts instead of 2D layouts? (Reviewer YpWR)

We note that state-of-the-art text-to-3D models (GraphDreamer, GALA3D) cannot generate reasonable results when given poorly initialized 3D layouts (**Fig. 5, 20** in our paper).

We use 2D layouts as guidance for the following reasons:
- Large language models (LLMs) are trained for general-purpose tasks and often struggle with reasoning, leading to inaccurate spatial relationships. Specifically, the 3D bounding boxes generated by LLMs tend to be imprecise and of low quality, which creates challenges in the subsequent 3D generation process. For visualization, we provide examples of 3D bounding boxes generated by LLMs in the rebuttal, please refer to **Fig. 18** and **Appendix D.1** in the manuscript.
- In contrast, 2D layout generation models are specifically designed for this task and produce much more accurate initial layouts. By first generating a 2D layout, we break down the complex problem into two more manageable subproblems: *2D layout generation and 3D lifting*. As a result, as demonstrated by the outcomes in the main paper, our framework generates higher-quality 3D scenes in a shorter amount of time.
- Finally, 2D layouts are much easier for applications like customization and interactive editing by users compared to 3D layouts.

---

### Meta-Review · Area_Chair_tQwb · 2024-12-22

**Metareview:**

The paper tackles the problem of controllable 3D scene generation. The proposed pipeline starts with 2D layout generation and then progresses through 3D instance reconstruction and arrangement. Their key insight is to decompose the original complex problem into a series of simpler and modular tasks. Then, by exploiting state-of-the-art modules for each stage, the proposed method is able to produce plausible 3D reconstructions with reasonable scene layouts. On the negative side, the reviewers are concerned that similar pipelines have already existed in the past literature and there are not sufficient and comprehensive evaluations/comparisons. The reviews were on the fence. During the discussion phase, the authors have attempted (and managed) to address most of the concerns. While there is still a debate going on regarding whether the novelty is enough and whether the work is incremental (ie, switching from generating 3D layout from LLM to producing 2D layout and lift it to 3D), considering the fact the authors incorporated quite a bit of additional results and showcased superior performance during the rebuttal, the ACs agree that the paper is marginally above the acceptance bar and decide to recommend acceptance. The authors are strongly encouraged to incorporate the feedback from the reviewers into their final camera ready version.

**Additional Comments On Reviewer Discussion:**

The authors provided additional experimental results per the reviewers' request and clarified several technical details.

---

### Decision · Program_Chairs · 2025-01-22

Accept (Poster)